# Exploring the combined impact of hepatitis B antibody status and systemic immune-inflammation index on mortality risk: A population-based study

Di Zeng[1,2], Shaofeng Wang[1,2], Nansheng Cheng[1,2], Bei Li[1,2], Xianze Xiong[1,2]*, Jiong Lu[1,2]*

**1** Division of Biliary Surgery, Department of General Surgery, West China Hospital, Sichuan University, Chengdu, Sichuan, China, **2** Research Center for Biliary Diseases, West China Hospital, Sichuan University, Chengdu, Sichuan, China

* xiongxz@163.com (XX); lujiong@scu.edu.cn (JL)

## Abstract

### Background

Chronic hepatitis B virus (HBV) infection is a significant global health issue, leading to liver-related morbidity and mortality. The systemic immune-inflammation index (SII), a marker of systemic inflammation and immune response, may predict disease outcomes. However, its role in HBV infection and its relationship with HBV surface antibody (HBsAb) status is not well understood. This study investigates the association between SII, HBsAb status, and their combined effects on all-cause and cardiovascular disease (CVD) mortality.

### Methods

We analyzed data from 43,539 participants in the National Health and Nutrition Examination Survey (NHANES), categorizing them into four groups based on SII and HBsAb status: high/low SII with HBsAb-negative/positive. Mortality outcomes were assessed using Cox proportional hazards models adjusted for age, sex, race/ethnicity, BMI, and comorbidities.

### Results

In the analysis of 43,539 participants, the fully adjusted model revealed that SII was significantly associated with both all-cause mortality (HR = 1.138, p < 0.001) and cardiovascular disease mortality (HR = 1.402, p < 0.0001), indicating that higher SII independently increases the risk of both outcomes. While the crude model showed a protective effect of HBV surface antibody on all-cause mortality (HR = 0.491, p < 0.0001) and cardiovascular disease mortality (HR = 0.478, p < 0.0001), this effect diminished after full adjustment. Additionally, the combined effect of SII and HBV

**Data availability statement:** This study utilized data from the National Health and Nutrition Examination Survey (NHANES), a publicly available dataset provided by the Centers for Disease Control and Prevention (CDC). All data were anonymized prior to analysis, and NHANES participants provided informed consent at the time of data collection. The NHANES protocol is reviewed and approved by the National Center for Health Statistics (NCHS) Research Ethics Review Board, and all procedures are conducted in accordance with the Declaration of Helsinki (https://www.cdc.gov/nchs/nhanes/index.htm).

**Funding:** This work was supported by the Sichuan Province Science and Technology Department Social Development Key Projects (No. 2023YFS0302), the Sichuan Science and Technology Program (Nos. 2024NSFSC0642 and 2024NSFSC0755), the Sichuan Provincial Department of Science and Technology Central-Guided Local Science and Technology Development Program (No. 2023ZYD0074), and the Sichuan University Innovation Research Project (No. 2023SCUH0041).

**Competing interests:** The authors have declared that no competing interests exist.

**Abbreviations:** Anti-HBs, Hepatitis B Surface Antibody; BMI, Body Mass Index; CVD, Cardiovascular Disease; HBV, Hepatitis B Virus; HCC, Hepatocellular Carcinoma; NAFLD, Non-Alcoholic Fatty Liver Disease; SII, Systemic Immune-Inflammation Index; TCGA, The Cancer Genome Atlas; OS, Overall Survival; DFS, Disease-Free Survival; CDC, Centers for Disease Control and Prevention; HR, Hazard Ratio; CI, Confidence Interval; WHO, World Health Organization; PLR, Platelet-to-Lymphocyte Ratio; NLR, Neutrophil-to-Lymphocyte Ratio; RBC, Red Blood Cell; WBC, White Blood Cell; CRP, C-Reactive Protein

surface antibody on both mortality outcomes remained significant in the fully adjusted model (HR = 1.402, p < 0.0001).

## Conclusion

Higher SII is independently associated with increased risks of all-cause and cardiovascular disease mortality. The protective effect of HBV surface antibody on mortality diminished after adjustment for confounders. The combined effect of SII and HBV surface antibody on mortality highlights the complex interaction between inflammation and immune response in chronic HBV infection. SII may serve as a useful predictor of long-term health risks in HBV-infected individuals.

## 1. Introduction

Hepatitis B virus (HBV) infection remains a major global health concern [1], with an estimated 250 million people living with chronic HBV worldwide [2]. The World Health Organization (WHO) has highlighted HBV as a leading cause of liver-related morbidity and mortality, contributing significantly to the development of liver cirrhosis, liver failure, and hepatocellular carcinoma (HCC) [3]. Despite the availability of effective vaccines, which have reduced new HBV infections globally, chronic HBV remains endemic in regions such as Asia and sub-Saharan Africa [4]. Chronic HBV infection can result in a range of adverse clinical outcomes, which are influenced by multiple factors such as the immune status of the host, viral load, and comorbidities including non-alcoholic fatty liver disease (NAFLD) and metabolic syndrome [5]. These factors not only complicate the management of HBV but also contribute to an increased risk of liver fibrosis, cirrhosis, and the eventual development of HCC [6]. Understanding the mechanisms that drive HBV progression and the impact of various biomarkers is essential for predicting disease outcomes and tailoring effective treatment strategies.

One such promising biomarker is the systemic immune-inflammation index (SII), which has garnered attention for its ability to predict prognosis in a variety of cancers, including HCC [7]. The SII is calculated as the product of platelet count and neutrophil count, divided by the lymphocyte count, reflecting a balance between inflammation and immune response [8]. A higher SII has been associated with a pro-tumor inflammatory state, characterized by elevated neutrophils and platelets, alongside a decrease in lymphocytes [9]. This inflammatory imbalance is believed to compromise immune surveillance, thus facilitating tumor progression [10]. Recent studies have linked high SII with poorer overall survival (OS) and disease-free survival (DFS) across several cancer types, including colorectal, gastric, and esophageal cancers [11–13]. However, its role in HBV infection, particularly in relation to HBV antibody status, remains underexplored.

HBV antibody status plays a critical role in determining the immune response to the virus [14]. Individuals who are positive for hepatitis B surface antibody (anti-HBs) may be more likely to maintain a more controlled immune response to HBV,

which could potentially influence the systemic inflammatory status in these individuals [15]. This might manifest as a comparatively lower systemic immune-inflammation index (SII), though this relationship is complex and could be influenced by various factors. On the other hand, those who are negative for HBV antibodies may experience a different immune dynamic, which could be associated with a more pronounced inflammatory state, possibly reflected in a higher SII [16].

The potential differences in immune profiles and inflammatory responses between HBV antibody-positive and -negative individuals require further investigation, as they may provide valuable insights into the immune dynamics of HBV infection and its progression. Our research suggests that variations in SII could have important implications for disease outcomes. A higher SII in HBV-positive individuals might indicate a more severe inflammatory state, contributing to liver fibrosis and hepatocellular carcinoma, while a lower SII in antibody-positive individuals may reflect better immune control and liver function. By exploring these associations, our study aims to deepen the understanding of HBV pathophysiology and inform the development of personalized therapeutic approaches, including optimized antiviral and immunomodulatory treatments for HBV patients.

## 2. Materials and methods

### 1. Data source

Data for this study were obtained from the National Health and Nutrition Examination Survey (NHANES), a nationally representative survey conducted by the Centers for Disease Control and Prevention (CDC) in the United States. The NHANES collects comprehensive data on the health, nutrition, and socioeconomic status of the civilian, non-institutionalized population. In this study, data from 2010 years to 2020 years were utilized to examine the relationship between hepatitis B virus (HBV) antibody status and the systemic immune-inflammation index (SII). The NHANES datasets are publicly available for research purposes through the CDC website (https://www.cdc.gov/nchs/nhanes/index.htm).

### 2. Study population

A total of 43539 participants from the selected NHANES cycles were included in this study. The inclusion criteria were individuals aged 18 years and older with available data on HBV antibody status and SII-related biomarkers (platelet count, neutrophil count, and lymphocyte count). Additionally, participants with complete survival data, as well as detailed information on covariates (such as age, sex, race/ethnicity, BMI, smoking history, diabetes, and hypertension), were included to ensure robust analysis. The exclusion criteria were as follows: (1) participants with missing data on HBV antibody testing, (2) individuals without information on SII biomarkers (platelet count, neutrophil count, and lymphocyte count), (3) those lacking critical covariate data, and (4) individuals with incomplete survival data, as accurate survival analysis requires complete and continuous follow-up information. **Participants with missing data on key variables—including HBV antibody testing, SII-related biomarkers (platelet, neutrophil, lymphocyte counts), covariates (age, sex, BMI, comorbidities), or survival outcomes—were excluded using listwise deletion. This method ensured that all models were run on a consistent and complete dataset, thereby improving the comparability and validity of the statistical results.** This ensured that the analysis of the relationship between SII and HBV antibody status could be performed with minimal bias and more reliable results.

### 3. HBV infection and antibody status

HBV antibody status was determined based on the detection of hepatitis B surface antibody (anti-HBs) in serum samples. Participants were classified as HBV antibody-positive (indicating past infection or successful vaccination) or HBV antibody-negative (indicating no detectable immunity or active infection). Serum samples were processed and stored following standard protocols by CDC-certified laboratories.

### 4. Systemic Immune–Inflammation Index (SII)

The SII was calculated using the following formula: SII = platelet count × neutrophil count/ lymphocyte count. A higher SII value indicates a pro-tumor inflammatory state characterized by increased neutrophils and platelets and decreased lymphocytes.

### 5. Statistical analysis

Descriptive statistics were calculated for all included variables. Continuous variables were expressed as means ± standard deviations, while categorical variables were presented as frequencies and percentages. Differences in clinical characteristics between HBV antibody-positive and negative groups were evaluated using independent t-tests for continuous variables and chi-square tests for categorical variables. To assess the relationship between SII and HBV antibody status, logistic regression analysis was employed. Three models were constructed: the unadjusted model (without covariate adjustment), the minimally adjusted model (adjusting for age, sex, race/ethnicity, BMI, smoking, diabetes, and hypertension), and the fully adjusted model (adjusting for all covariates, including dietary factors and other health behaviors). The association between SII and clinical outcomes, including liver function, was further evaluated using survival analysis techniques, such as Kaplan-Meier curves and Cox proportional hazards models.

### 6. Ethics statement placement

This study utilized data from the National Health and Nutrition Examination Survey (NHANES), a publicly available dataset provided by the Centers for Disease Control and Prevention (CDC). All data were anonymized prior to analysis, and NHANES participants provided informed consent at the time of data collection. The NHANES protocol is reviewed and approved by the National Center for Health Statistics (NCHS) Research Ethics Review Board, and all procedures are conducted in accordance with the Declaration of Helsinki.

## 3. Results

### 1. Patients' inclusion

This study included 43,539 patients, categorized into four groups based on systemic immune-inflammation index (SII) and hepatitis B surface antibody (HBsAb) status: high SII and HBsAb-negative (17,677), high SII and HBsAb-positive (4,090), low SII and HBsAb-negative (16,656), and low SII and HBsAb-positive (5,116). The mean age was 46.99 years, with older patients in the high SII groups (high SII/HBsAb-negative: 49.33 years) and younger patients in the low SII/HBsAb-positive group (39.80 years). Mean BMI was 28.80 kg/m², highest in the high SII/HBsAb-negative group (29.55 kg/m²) and lowest in the low SII/HBsAb-positive group (27.09 kg/m²). Lipid profiles showed higher triglycerides and total cholesterol in high SII groups, while HDL levels were higher in the low SII/HBsAb-positive group (1.42 mmol/L). Renal function, assessed by CKD-EPI eGFR, was lower in high SII groups (92.17 mL/min/1.73 m²) and higher in HBsAb-positive groups (100.73 mL/min/1.73 m²). Females predominated in the high SII/HBsAb-positive group (62.08%), while males were more common in the low SII/HBsAb-negative group (54.13%). Non-Hispanic Whites comprised the majority (69.81%), with higher diversity in the low SII/HBsAb-positive group. Education levels were higher in the high SII/HBsAb-positive group (69.63% above high school). Hypertension and CKD prevalence were higher in high SII groups, while COPD prevalence was lowest in the low SII/HBsAb-positive group (1.75%). In summary, high SII groups exhibited higher cardiometabolic risk profiles, whereas HBsAb-positive groups were younger with more favorable metabolic indicators. Details of inculded patients are shown in Table 1.

### 2. SII and all-cause mortality

The association between SII and all-cause mortality was assessed across quartiles of SII. In the crude model, individuals in the highest SII quartile (Q4) had a significantly higher risk of all-cause mortality (HR = 1.339, 95% CI: 1.224–1.464,

**Table 1. Baseline characteristics of the study population stratified by systemic immune-inflammation index (SII) and hepatitis B surface antibody (HBsAb) status.**

| Variables | Total | High SII and HBsAb (-) (17677) | High SII and HBsAb (+) (4090) | Low SII and HBsAb (-) (16656) | Low SII and HBsAb (+) (5116) | P value |
|---|---|---|---|---|---|---|
| Age | 46.99 (0.19) | 49.33 (0.23) | 39.48 (0.33) | 48.80 (0.24) | 39.80 (0.31) | < 0.0001 |
| BMI | 28.80 (0.06) | 29.55 (0.08) | 28.44 (0.15) | 28.62 (0.08) | 27.09 (0.12) | < 0.0001 |
| TG | 1.69 (0.01) | 1.77 (0.02) | 1.57 (0.03) | 1.73 (0.02) | 1.45 (0.02) | < 0.0001 |
| Total_cholesterol | 5.08 (0.01) | 5.14 (0.01) | 4.96 (0.02) | 5.10 (0.01) | 4.90 (0.02) | < 0.0001 |
| HDL | 1.38 (0.00) | 1.36 (0.01) | 1.42 (0.01) | 1.37 (0.01) | 1.42 (0.01) | < 0.0001 |
| LDL | 2.93 (0.01) | 2.97 (0.01) | 2.83 (0.02) | 2.95 (0.01) | 2.83 (0.02) | < 0.0001 |
| Albumin | 42.78 (0.04) | 42.44 (0.05) | 42.68 (0.08) | 43.00 (0.05) | 43.38 (0.07) | < 0.0001 |
| CKD_EPI_Scr_2009 | 94.26 (0.25) | 92.17 (0.30) | 100.73 (0.47) | 92.68 (0.28) | 100.73 (0.41) | < 0.0001 |
| SII | 491.30 (355.83, 688.27) | 684.47 (569.68, 872.00) | 665.13 (561.46, 834.62) | 354.32 (280.30, 417.59) | 345.00 (271.30, 410.63) | < 0.0001 |
| Sex |  |  |  |  |  | < 0.0001 |
| Female | 22225 (51.29) | 9480 (53.75) | 2504 (62.08) | 7650 (45.87) | 2591 (50.29) |  |
| Male | 21314 (48.71) | 8197 (46.25) | 1586 (37.92) | 9006 (54.13) | 2525 (49.71) |  |
| Eth1 |  |  |  |  |  | < 0.0001 |
| Mexican American | 7732 (8.07) | 3541 (8.44) | 550 (6.80) | 3109 (8.72) | 532 (5.85) |  |
| Non-Hispanic Black | 8650 (10.34) | 2331 (6.75) | 753 (9.25) | 4045 (12.87) | 1521 (15.80) |  |
| Non-Hispanic White | 19968 (69.81) | 9483 (75.08) | 1798 (66.47) | 6929 (67.44) | 1758 (61.79) |  |
| Other Hispanic | 3535 (5.44) | 1355 (5.18) | 359 (6.32) | 1393 (5.47) | 428 (5.47) |  |
| Other Race – Including Multi-Racial | 3654 (6.35) | 967 (4.55) | 630 (11.15) | 1180 (5.49) | 877 (11.09) |  |
| Edu |  |  |  |  |  | < 0.0001 |
| Above high school | 21987 (59.35) | 8303 (55.34) | 2518 (69.63) | 7964 (57.27) | 3202 (70.75) |  |
| High school or equivalent | 10075 (23.96) | 4391 (26.72) | 847 (19.26) | 3872 (24.18) | 965 (17.87) |  |
| Under high school | 11477 (16.69) | 4983 (17.95) | 725 (11.11) | 4820 (18.55) | 949 (11.38) |  |
| Family_income |  |  |  |  |  | 0.01 |
| High | 13698 (43.25) | 5371 (42.15) | 1411 (45.34) | 5208 (43.75) | 1708 (43.75) |  |
| Low | 13272 (21.02) | 5479 (21.07) | 1241 (22.27) | 5054 (20.57) | 1498 (21.16) |  |
| Medium | 16569 (35.73) | 6827 (36.78) | 1438 (32.38) | 6394 (35.68) | 1910 (35.09) |  |
| Marital |  |  |  |  |  | < 0.0001 |
| Married | 26465 (64.16) | 10792 (64.03) | 2291 (57.41) | 10532 (67.68) | 2850 (59.54) |  |
| Other | 17074 (35.84) | 6885 (35.97) | 1799 (42.59) | 6124 (32.32) | 2266 (40.46) |  |
| Alcohol user |  |  |  |  |  | < 0.0001 |
| Former | 7740 (14.47) | 3519 (16.75) | 549 (11.03) | 3016 (14.44) | 656 (9.69) |  |
| Heavy | 8647 (21.35) | 3486 (21.12) | 992 (26.35) | 3109 (19.79) | 1060 (22.64) |  |
| Mild | 14330 (35.66) | 5683 (34.76) | 1247 (32.77) | 5630 (37.12) | 1770 (36.72) |  |
| Moderate | 6459 (17.05) | 2455 (16.22) | 741 (19.44) | 2428 (16.67) | 835 (19.04) |  |
| Never | 6363 (11.47) | 2534 (11.15) | 561 (10.41) | 2473 (11.98) | 795 (11.91) |  |
| Smoke |  |  |  |  |  | < 0.0001 |
| Former | 10970 (24.98) | 4786 (26.52) | 812 (19.15) | 4382 (26.39) | 990 (20.35) |  |
| Never | 23470 (53.59) | 8966 (49.86) | 2407 (59.71) | 8960 (53.79) | 3137 (60.61) |  |
| Now | 9099 (21.42) | 3925 (23.62) | 871 (21.14) | 3314 (19.82) | 989 (19.04) |  |
| HBV_surface_antibody |  |  |  |  |  | < 0.0001 |
| Negative | 34333 (77.83) | 17677 (100.00) | 0 (0.00) | 16656 (100.00) | 0 (0.00) |  |
| Positive | 9206 (22.17) | 0 (0.00) | 4090 (100.00) | 0 (0.00) | 5116 (100.00) |  |

*(Continued)*

Table 1. (Continued)

| Variables | Total | High SII and HBsAb (-) (17677) | High SII and HBsAb (+) (4090) | Low SII and HBsAb (-) (16656) | Low SII and HBsAb (+) (5116) | P value |
|---|---|---|---|---|---|---|
| Hypertension | | | | | | < 0.0001 |
| No | 25203 (63.00) | 9591 (58.39) | 2757 (71.02) | 9321 (62.03) | 3534 (74.96) | |
| Yes | 18336 (37.00) | 8086 (41.61) | 1333 (28.98) | 7335 (37.97) | 1582 (25.04) | |
| CKD | | | | | | < 0.0001 |
| No | 35624 (85.98) | 13850 (83.34) | 3507 (89.12) | 13758 (86.29) | 4509 (91.38) | |
| Yes | 7915 (14.02) | 3827 (16.66) | 583 (10.88) | 2898 (13.71) | 607 (8.62) | |
| COPD | | | | | | < 0.0001 |
| No | 41714 (95.99) | 16679 (94.54) | 3972 (97.47) | 16054 (96.46) | 5009 (98.25) | |
| Yes | 1825 (4.01) | 998 (5.46) | 118 (2.53) | 602 (3.54) | 107 (1.75) | |

Note: Values are presented as mean, with standard error shown in parentheses (e.g., 46.99 (0.19) indicates a mean of 46.99 and a standard error of 0.19).

p < 0.0001) compared to those in the lowest quartile (Q1). After adjusting for age and sex (Model 1), the association remained significant (HR = 1.302, 95% CI: 1.207–1.405, p < 0.0001). Further adjustment for socio-economic and clinical factors in Models 2 and 3 did not substantially alter the results, with the HR for Q4 remaining elevated (HR = 1.318, 95% CI: 1.222–1.421, p < 0.0001 in Model 2 and HR = 1.138, 95% CI: 1.054–1.229, p < 0.001 in Model 3). This indicates that higher SII is independently associated with increased all-cause mortality risk. The detailed content above is provided in Table 2A.

### 3. SII and cardiovascular disease mortality

The association between SII and cardiovascular disease (CVD) mortality was evaluated across quartiles of SII. In the crude model, participants in the highest SII quartile (Q4) showed a significantly increased risk of cardiovascular mortality (HR = 1.575, 95% CI: 1.343–1.847, p < 0.0001) compared to those in the lowest quartile (Q1). After adjusting for age and sex (Model 1), the risk remained elevated (HR = 1.590, 95% CI: 1.367–1.851, p < 0.0001). Further adjustments for additional socio-economic and clinical factors in Models 2 and 3 did not change the association substantially, with the HR for Q4 remaining significant (HR = 1.655, 95% CI: 1.415–1.936, p < 0.0001 in Model 2 and HR = 1.402, 95% CI: 1.191–1.649, p < 0.0001 in Model 3). This suggests that higher SII is independently associated with increased cardiovascular mortality risk. The detailed content above is provided in Table 2B. The all-cause mortality, CVD death, and non-CVD death associated with the high and low SII groups are presented in Fig 1a–1c, respectively.

### 4. HBV surface antibody and all-cause mortality

For HBV surface antibody status, participants who were HBV antibody-positive had a significantly lower risk of all-cause mortality in the crude model (HR = 0.491, 95% CI: 0.441–0.547, p < 0.0001). However, after adjusting for age and sex in Model 1, the HR increased and became non-significant (HR = 0.909, 95% CI: 0.819–1.009, p = 0.072). Further adjustment in Models 2 and 3 continued to show no significant association (HR = 0.878, 95% CI: 0.788–0.978, p = 0.018 in Model 2 and HR = 0.906, 95% CI: 0.809–1.014, p = 0.086 in Model 3). This suggests that while HBV antibody positivity may initially appear protective, this effect is diminished after controlling for other factors. The detailed content above is provided in Table 3A.

**Table 2. Hazard ratios (HRs) and 95% confidence intervals (CIs) for (A) all-cause mortality across quartiles of systemic immune-inflammation index (SII) and (B) cardiovascular disease (CVD) mortality across quartiles of systemic immune-inflammation index (SII).**

nhs~permth_int, mortstat

| Character | SIIQ Crude model 95%CI | P | Model 1 95%CI | P | Model 2 95%CI | P | Model 3 95%CI | P |
|---|---|---|---|---|---|---|---|---|
| Q1 | ref | | ref | | ref | | ref | |
| Q2 | 0.883 (0.801, 0.973) | 0.012 | 0.904 (0.826, 0.990) | 0.029 | 0.944 (0.858, 1.039) | 0.241 | 0.922 (0.837, 1.015) | 0.096 |
| Q3 | 0.962 (0.876, 1.057) | 0.423 | 0.966 (0.882, 1.057) | 0.452 | 1.011 (0.923, 1.108) | 0.809 | 0.953 (0.868, 1.045) | 0.304 |
| Q4 | 1.339 (1.224, 1.464) | <0.0001 | 1.302 (1.207, 1.405) | <0.0001 | 1.318 (1.222, 1.421) | <0.0001 | 1.138 (1.054, 1.229) | <0.001 |
| p for trend (character2integer) | | <0.0001 | | <0.0001 | | <0.0001 | | <0.0001 |
| p for trend (Median value) | | <0.0001 | | <0.0001 | | <0.0001 | | <0.0001 |

SIIQ
Crudel model: SIIQ
Model 1: SIIQ, age, sex
Model 2: SIIQ, age, sex, eth1, Family_income, edu, marital
Model 3: SIIQ, age, sex, eth1, marital, Family_income, edu, BMI, TG, HDL, total_cholesterol, LDL, HDL, TG, Albumin, CKD_EPI_Scr_2009, smoke, alcohol.user, CKD, Hypertension, COPD

Subset (nhs, ucod_leading %in% c (Cerebrovascular, heart, no))~permth_int, mortstat

| Character | SIIQ Crude model 95%CI | P | Model 1 95%CI | P | Model 2 95%CI | P | Model 3 95%CI | P |
|---|---|---|---|---|---|---|---|---|
| Q1 | ref | | ref | | ref | | ref | |
| Q2 | 0.978 (0.820, 1.165) | 0.801 | 1.017 (0.867, 1.192) | 0.837 | 1.091 (0.926, 1.284) | 0.298 | 1.031 (0.869, 1.224) | 0.724 |
| Q3 | 1.063 (0.912, 1.240) | 0.435 | 1.091 (0.939, 1.269) | 0.255 | 1.170 (1.004, 1.364) | 0.045 | 1.087 (0.927, 1.273) | 0.304 |
| Q4 | 1.575 (1.343, 1.847) | <0.0001 | 1.590 (1.367, 1.851) | <0.0001 | 1.655 (1.415, 1.936) | <0.0001 | 1.402 (1.191, 1.649) | <0.0001 |
| p for trend (character2integer) | | <0.0001 | | <0.0001 | | <0.0001 | | <0.0001 |
| p for trend (Median value) | | <0.0001 | | <0.0001 | | <0.0001 | | <0.0001 |

SIIQ
Crudel model: SIIQ
Model 1: SIIQ, age, sex
Model 2: SIIQ, age, sex, eth1, Family_income, edu, marital
Model 3: SIIQ, age, sex, eth1, marital, Family_income, edu, BMI, TG, HDL, total_cholesterol, LDL, HDL, TG, Albumin, CKD_EPI_Scr_2009, smoke, alcohol.user, CKD, Hypertension, COPD

## 5. HBV surface antibody and cardiovascular disease mortality

For HBV surface antibody status, the crude model indicated a significant protective effect of being HBV antibody-positive, with a lower hazard ratio for cardiovascular mortality (HR = 0.478, 95% CI: 0.397–0.576, p < 0.0001) compared to the antibody-negative group. However, after adjusting for age and sex in Model 1, the HR increased and became non-significant (HR = 0.981, 95% CI: 0.827–1.164, p = 0.829). Further adjustments in Models 2 and 3 continued to show a non-significant association (HR = 0.920, 95% CI: 0.773–1.095, p = 0.347 in Model 2 and HR = 0.946, 95% CI: 0.788–1.135, p = 0.551 in Model 3). This suggests that HBV antibody positivity may have an initial protective effect, but this association is attenuated after adjusting for other factors. The detailed content above is provided in Table 3B. The all-cause mortality, CVD death, and non-CVD death associated with HBV antibody-negative and -positive groups are presented in Fig 2a–2c, respectively.

## 6. Combined effect of SII and HBV surface antibody on all-cause mortality

When examining the combined effect of SII and HBV surface antibody status on all-cause mortality, a strong trend emerged for individuals with high SII and HBV antibody-negative status. In the crude model, the HR for this group was

(a)    All_cause_mortality

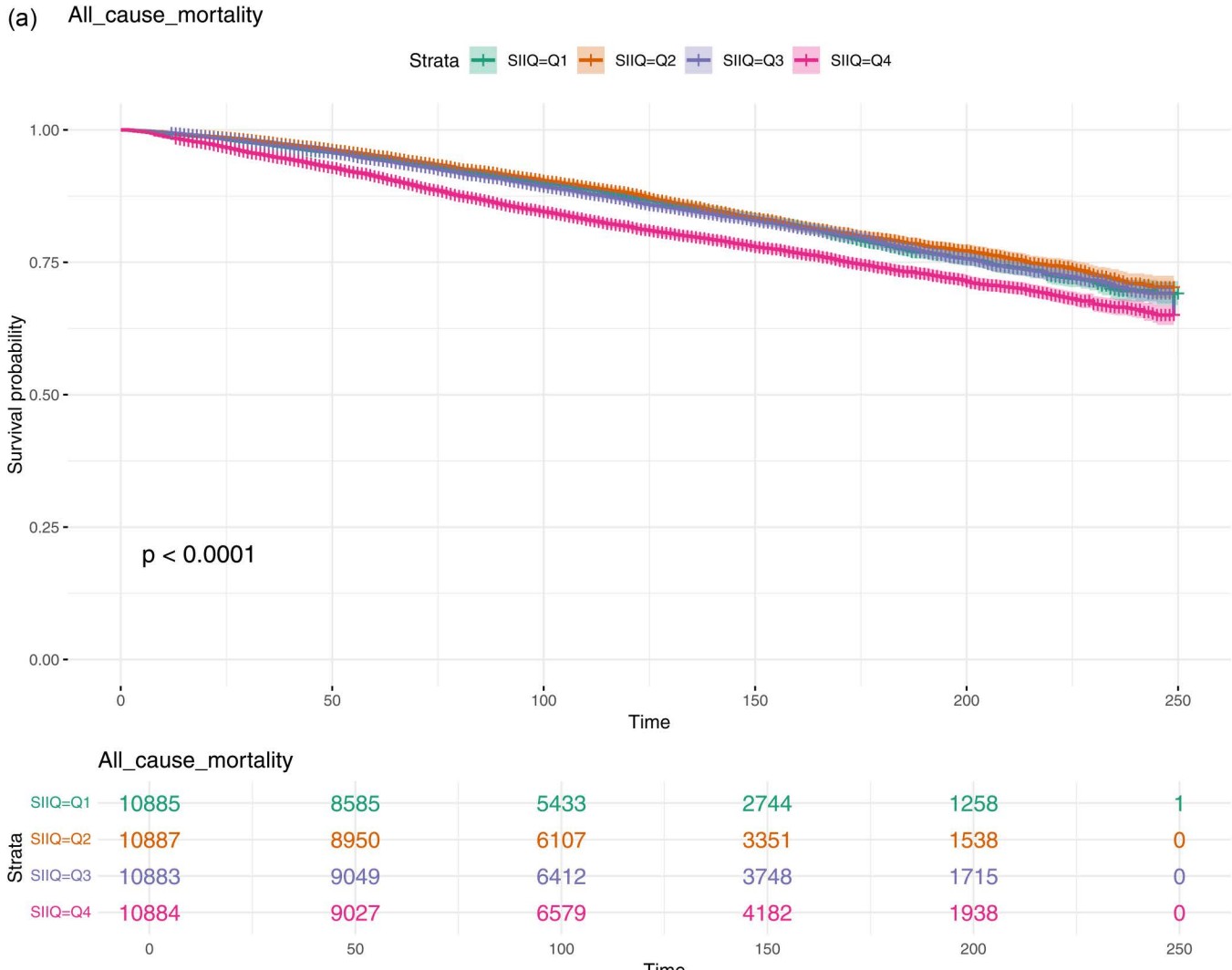

(b)    CVD_cause_mortality

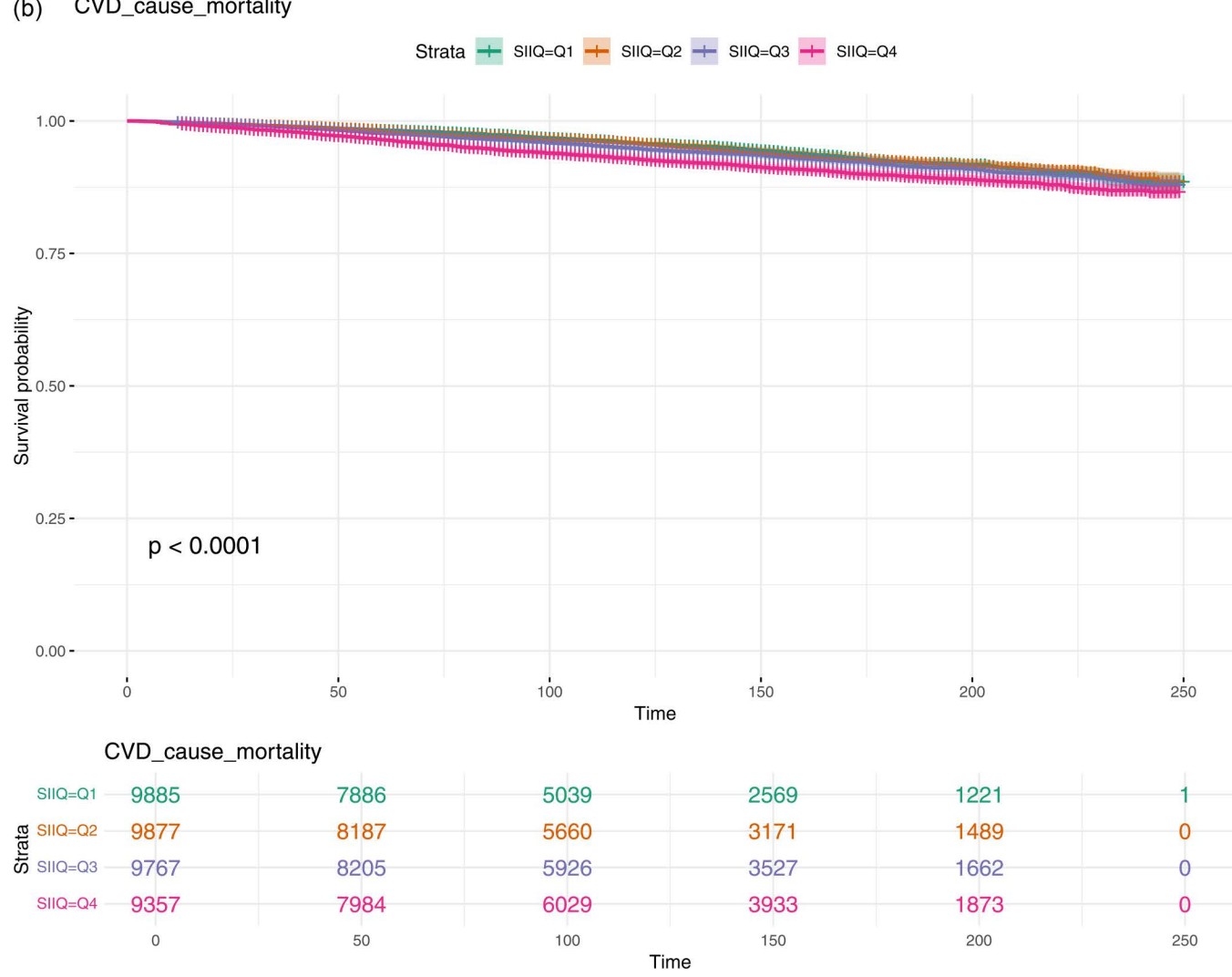

1.575 (95% CI: 1.343–1.847, p < 0.0001), and it remained significant after adjusting for age and sex (HR = 1.590, 95% CI: 1.367–1.851, p < 0.0001). Further adjustments in Models 2 and 3 confirmed that high SII continued to be associated with significantly higher all-cause mortality risk (HR = 1.655, 95% CI: 1.415–1.936, p < 0.0001 in Model 2 and HR = 1.402, 95% CI: 1.191–1.649, p < 0.0001 in Model 3). The detailed content above is provided in Table 4A.

## 7. Combined effect of SII and HBV surface antibody on cardiovascular disease mortality

When examining the combined effect of SII and HBV surface antibody status on cardiovascular mortality, a clear trend emerged. Participants with high SII and HBV antibody-negative status exhibited the worst survival outcomes. In the crude model, the HR for this group was 1.575 (95% CI: 1.343–1.847, p < 0.0001), and after adjusting for age and sex, the HR remained significant (HR = 1.590, 95% CI: 1.367–1.851, p < 0.0001). Further adjustments in Models 2 and 3 confirmed the strong association between high SII and poor cardiovascular outcomes, with the HR for Q4 consistently higher than for the lower SII groups. This indicates that while HBV antibody positivity might provide some protection, the risk conferred by high SII on cardiovascular mortality remains significant. The detailed content above is provided in Table 4B. The all-cause

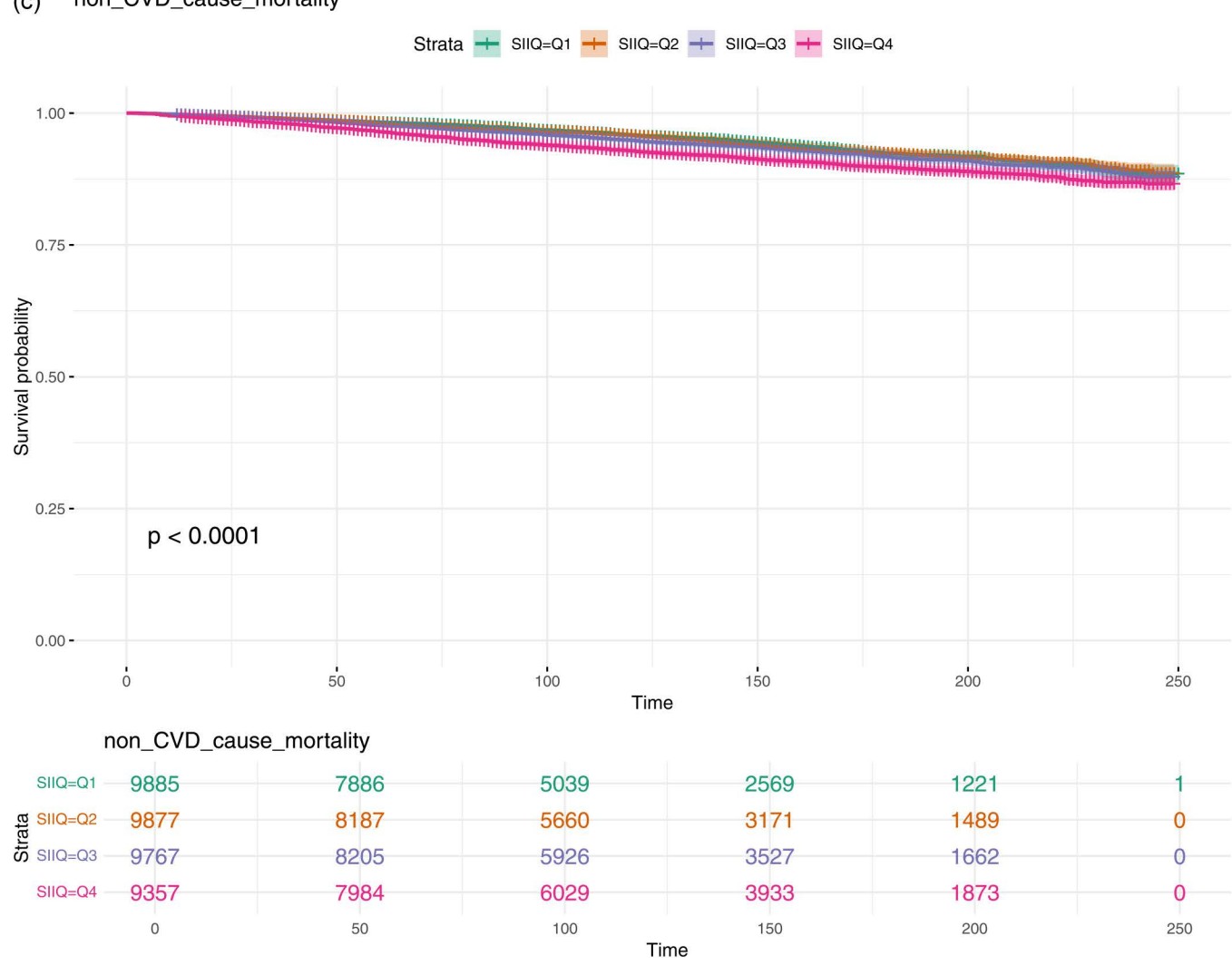

(c) non_CVD_cause_mortality

**Fig 1. (a) All-cause mortality associated with high and low systemic immune-inflammation index (SII) groups; (b) Cardiovascular disease (CVD) mortality associated with high and low systemic immune-inflammation index (SII) groups; (c) Non-cardiovascular disease (non-CVD) mortality associated with high and low systemic immune-inflammation index (SII) groups.**

mortality, CVD death, and non-CVD death associated with the combined SII and HBV antibody status groups are presented in Fig 3a–3c, respectively.

## 4. Discussion

In this study, we examined the relationship between the systemic immune-inflammation index (SII), HBV surface antibody status, and their combined effects on all-cause and cardiovascular disease (CVD) mortality. Our results suggest that both SII and HBV surface antibody status are significant predictors of mortality, with high SII associated with increased risks, particularly for all-cause and cardiovascular mortality. However, the protective effect of HBV surface antibody positivity diminished after adjusting for other clinical and sociodemographic factors [17], indicating that while there may be an initial protective effect, it is not robust when considering other health factors [18,19].

**Table 3. Hazard ratios (HRs) and 95% confidence intervals (CIs) for (A) all-cause mortality by hepatitis B surface antibody (HBsAb) status and (B) cardiovascular disease (CVD) mortality by hepatitis B surface antibody (HBsAb) status.**

**nhs~~permth_int, mortstat**

| Character | HBV_surface_antibody | | | | | | | |
|---|---|---|---|---|---|---|---|---|
| | Crude model | | Model 1 | | Model 2 | | Model 3 | |
| | 95%CI | P | 95%CI | P | 95%CI | P | 95%CI | P |
| Negative | ref | | ref | | ref | | ref | |
| Positive | 0.491 (0.441, 0.547) | <0.0001 | 0.909 (0.819, 1.009) | 0.072 | 0.878 (0.788, 0.978) | 0.018 | 0.906 (0.809, 1.014) | 0.086 |

**HBV_surface_antibody**

Crudel model: HBV_surface_antibody

Model 1: HBV_surface_antibody, age, sex

Model 2: HBV_surface_antibody, age, sex, eth1, Family_income, edu, marital

Model 3: HBV_surface_antibody, age, sex, eth1, marital, Family_income, edu, BMI, total_cholesterol, LDL, HDL, TG, Albumin, CKD_EPI_Scr_2009, smoke, alcohol.user, CKD, Hypertension, COPD

**Subset (nhs, ucod_leading %in% c(Cerebrovascular, heart, no))~~permth_int, mortstat**

| Character | HBV_surface_antibody | | | | | | | |
|---|---|---|---|---|---|---|---|---|
| | Crude model | | Model 1 | | Model 2 | | Model 3 | |
| | 95%CI | P | 95%CI | P | 95%CI | P | 95%CI | P |
| Negative | ref | | ref | | ref | | ref | |
| Positive | 0.478 (0.397, 0.576) | <0.0001 | 0.981 (0.827, 1.164) | 0.829 | 0.920 (0.773, 1.095) | 0.347 | 0.946 (0.788, 1.135) | 0.551 |

**HBV_surface_antibody**

Crudel model: HBV_surface_antibody

Model 1: HBV_surface_antibody, age, sex

Model 2: HBV_surface_antibody, age, sex, eth1, Family_income, edu, marital

Model 3: HBV_surface_antibody, age, sex, eth1, marital, Family_income, edu, BMI, total_cholesterol, LDL, HDL, TG, Albumin, CKD_EPI_Scr_2009, smoke, alcohol.user, CKD, Hypertension, COPD

Our findings confirm previous research highlighting the importance of inflammatory markers in predicting mortality. SII, which integrates platelet, neutrophil, and lymphocyte counts, serves as a novel marker of systemic inflammation and immune status. Elevated SII has been shown to correlate with worse outcomes in various diseases, including cancer and cardiovascular conditions [20]. In our study, participants with higher SII (particularly in the fourth quartile) had significantly higher risks of both all-cause and CVD mortality, even after adjusting for a wide range of potential confounders. This supports the hypothesis that SII could serve as an effective biomarker for identifying inflammation-driven mortality risks in the general population [21].

HBV surface antibody positivity has traditionally been viewed as a marker of immunity against hepatitis B virus infection [22]. Initially, our results showed that HBV antibody positivity was associated with lower risks of all-cause and cardiovascular mortality in the crude models. However, after adjusting for sociodemographic factors, comorbidities, and lifestyle factors, this protective association diminished and became non-significant in the final models. These findings suggest that while HBV surface antibodies may confer some protection against infection-related outcomes, their role in broader mortality outcomes is less pronounced once other health factors are accounted for. This aligns with previous studies that

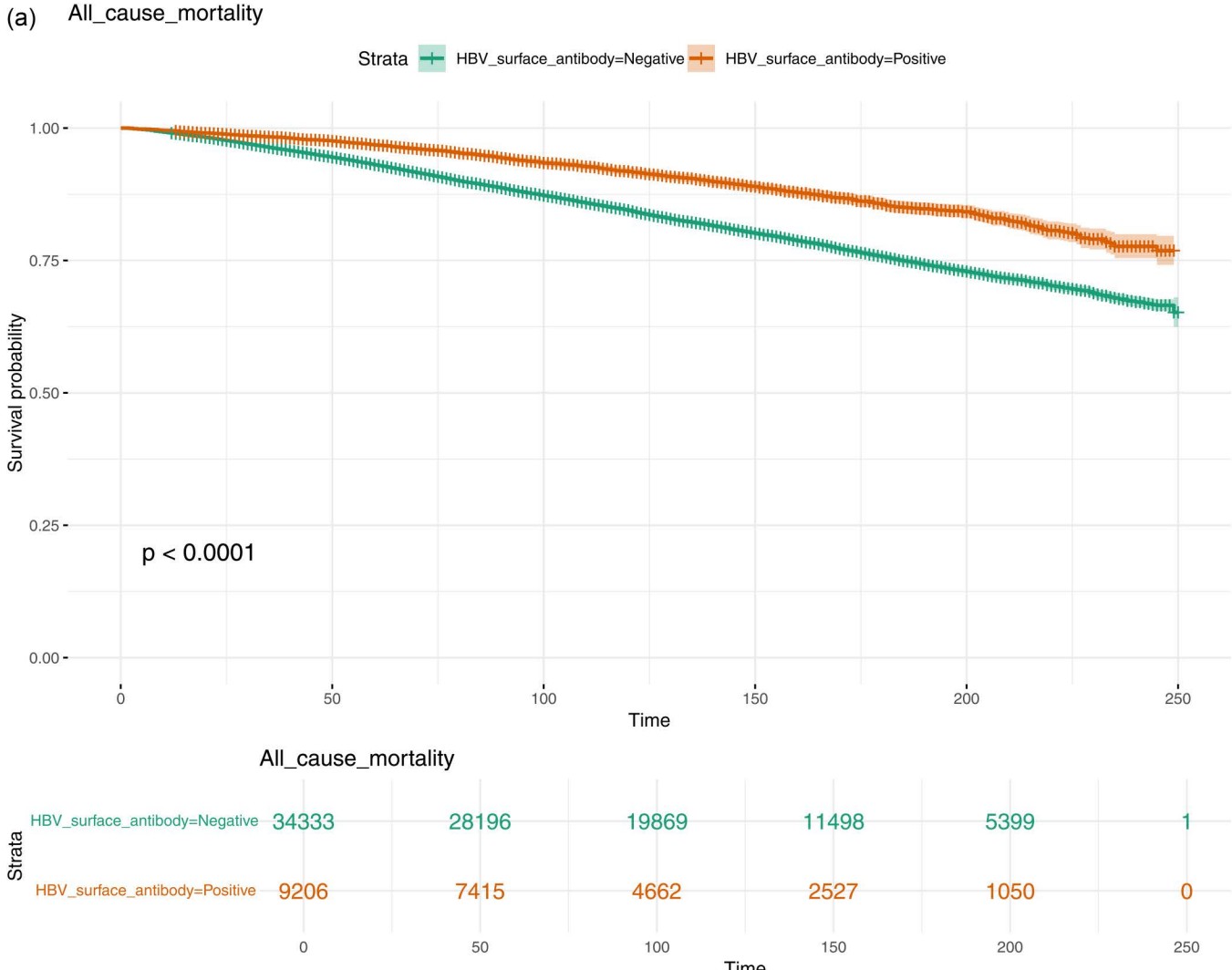

(a) All_cause_mortality

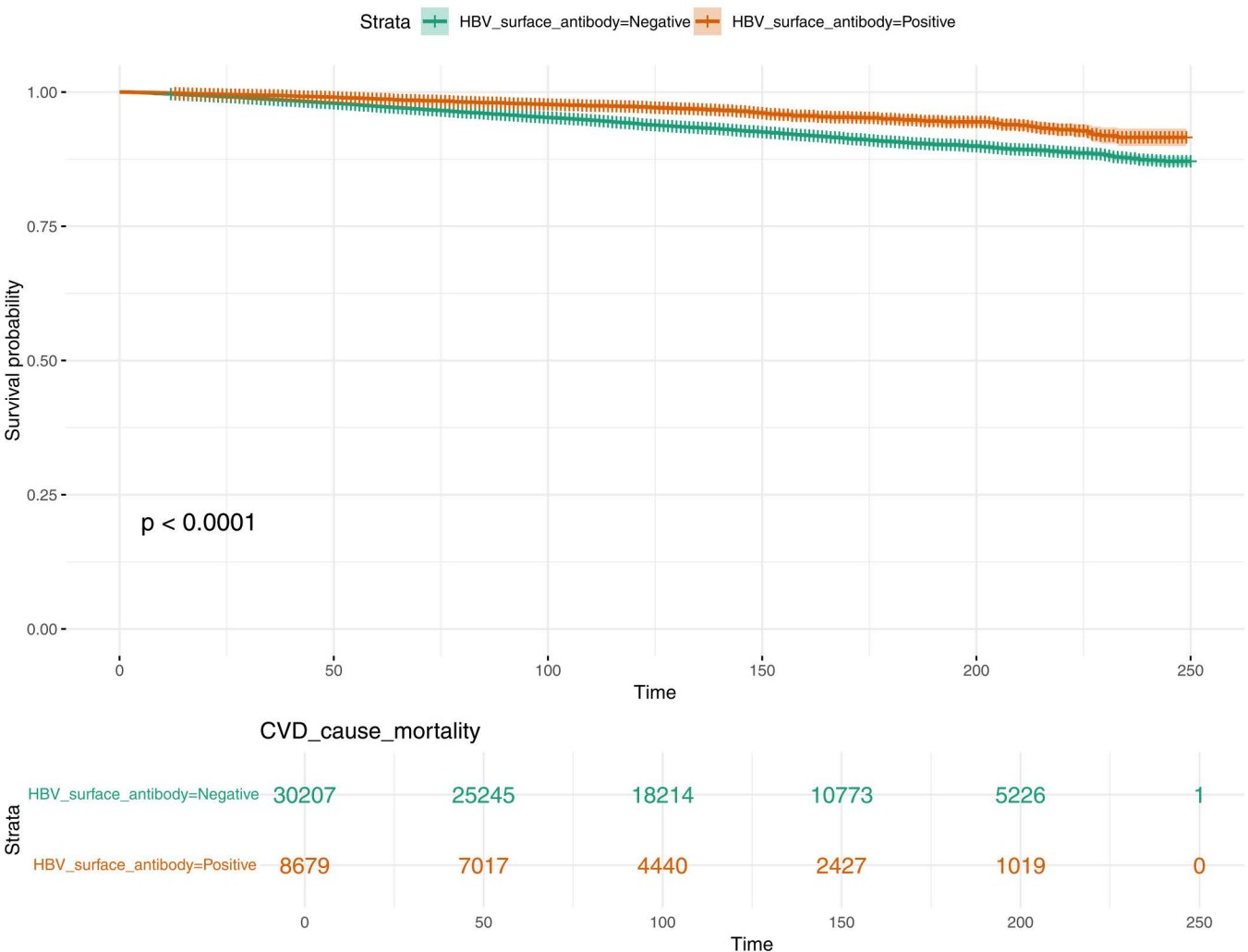

(b) CVD_cause_mortality

suggest the protective effect of HBV antibodies may be limited to preventing acute infections, rather than influencing chronic disease outcomes like cardiovascular events [23,24].

The combined effect of SII and HBV surface antibody status revealed a more complex relationship [25]. Participants with high SII and HBV antibody-negative status had the highest mortality risks across all models, emphasizing the detrimental effects of both systemic inflammation and the absence of immunity to HBV on health outcomes. This suggests that individuals with high levels of inflammation and no prior immunity to HBV may be at particularly elevated risk for poor outcomes. Conversely, those with high SII but HBV antibody-positive status had a moderately lower risk, suggesting that prior vaccination or immunity could attenuate the negative effects of systemic inflammation [18].

This finding underscores the importance of considering both immune status and inflammation when evaluating mortality risk [26]. While SII provides insight into systemic inflammation, HBV surface antibody status reflects an individual's immune history and protective status against viral infections [27]. By combining these two markers, we gain a more nuanced understanding of how immune and inflammatory responses contribute to overall mortality, including

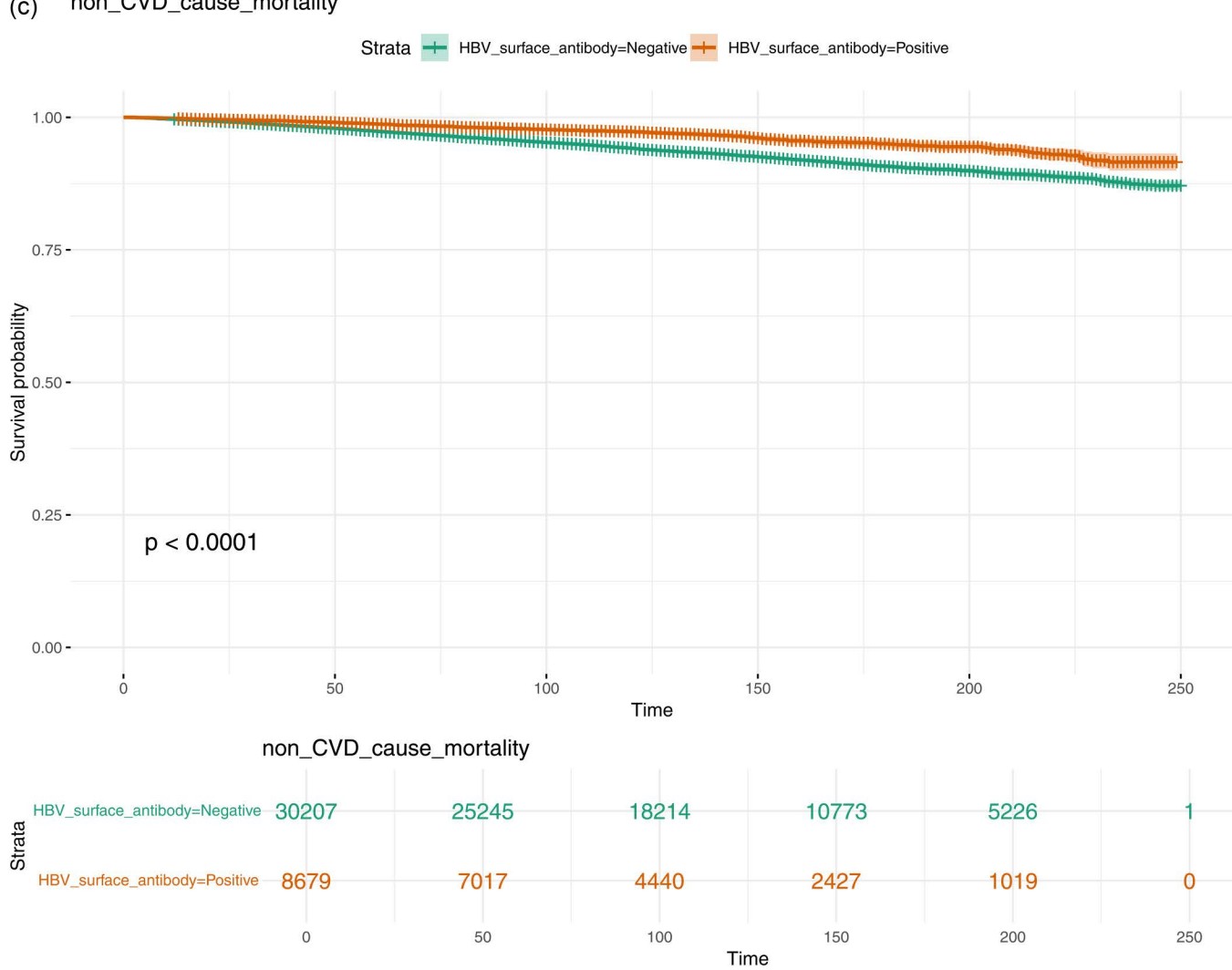

**Fig 2. (a) All-cause mortality associated with hepatitis B surface antibody (HBsAb)-negative and HBsAb-positive groups; (b) Cardiovascular disease (CVD) mortality associated with hepatitis B surface antibody (HBsAb)-negative and HBsAb-positive groups; (c) Non-cardiovascular disease (non-CVD) mortality associated with hepatitis B surface antibody (HBsAb)-negative and HBsAb-positive groups.**

cardiovascular outcomes. **It is** important to acknowledge that the proposed mechanisms linking elevated SII and the attenuation of the protective effect of HBsAb remain speculative. While our findings suggest a potential interplay between systemic inflammation and immune protection conferred by HBsAb, the underlying biological pathways have yet to be fully elucidated. Future mechanistic studies are warranted to explore how immune status and inflammatory responses interact to influence long-term health outcomes, particularly in HBV-affected populations. Moreover, it is important to consider that HBsAb positivity in this study likely includes both vaccine-induced and infection-induced immunity, which may represent biologically distinct immune profiles. Individuals with resolved HBV infection may exhibit different levels of residual inflammation or liver damage compared to those with vaccine-induced immunity, potentially influencing their SII levels and mortality risk. Without differentiation, these underlying differences may confound the interpretation of our findings. Future studies with serological markers such as anti-HBc are needed to disentangle these effects and refine the understanding of how HBV immune history interacts with systemic inflammation.

**Table 4. Hazard ratios (HRs) and 95% confidence intervals (CIs) for (A) all-cause mortality by combined systemic immune-inflammation index (SII) and hepatitis B surface antibody (HBsAb) status and (B) cardiovascular disease (CVD) mortality by combined systemic immune-inflammation index (SII) and hepatitis B surface antibody (HBsAb) status.**

**group_hbv_SII**

| Subset (nhs, ucod_leading %in% c (Cerebrovascular, heart, no))~~permth_int, mortstat | Crude model | | Model 1 | | Model 2 | | Model 3 | |
|---|---|---|---|---|---|---|---|---|
| Character | 95%CI | P | 95%CI | P | 95%CI | P | 95%CI | P |
| Low SII and HBsAg(+) | ref | | ref | | ref | | ref | |
| Low SII and HBsAg (-) | 2.123 (1.677, 2.688) | <0.0001 | 1.083 (0.853, 1.377) | 0.512 | 1.167 (0.915, 1.488) | 0.214 | 1.155 (0.903, 1.479) | 0.251 |
| High SII and HBsAg (-) | 2.745 (2.144, 3.515) | <0.0001 | 1.414 (1.099, 1.821) | 0.007 | 1.541 (1.200, 1.980) | <0.001 | 1.386 (1.073, 1.791) | 0.012 |
| High SII and HBsAg (+) | 1.368 (0.982, 1.906) | 0.064 | 1.527 (1.120, 2.083) | 0.007 | 1.552 (1.152, 2.091) | 0.004 | 1.448 (1.068, 1.964) | 0.017 |
| p for trend(character2integer) | | <0.0001 | | <0.0001 | | <0.0001 | | <0.001 |

Group_hbv_SII

Crudel model: group_hbv_SII

Model 1: group_hbv_SII, age, sex

Model 2: group_hbv_SII, age, sex, eth1, Family_income, edu, marital

Model 3: group_hbv_SII, age, sex, eth1, marital, Family_income, edu, BMI, total_cholesterol, LDL, HDL, TG, Albumin, CKD_EPI_Scr_2009, smoke, alcohol.user, CKD, Hypertension, COPD

**group_hbv_SII**

| Subset (nhs, ucod_leading %in% c (Cerebrovascular, heart, no))~~permth_int, mortstat | Crude model | | Model 1 | | Model 2 | | Model 3 | |
|---|---|---|---|---|---|---|---|---|
| Character | 95%CI | P | 95%CI | P | 95%CI | P | 95%CI | P |
| Low SII and HBsAg (+) | ref | | ref | | ref | | ref | |
| Low SII and HBsAg (-) | 2.123 (1.677, 2.688) | <0.0001 | 1.083 (0.853, 1.377) | 0.512 | 1.167 (0.915, 1.488) | 0.214 | 1.155 (0.903, 1.479) | 0.251 |
| High SII and HBsAg (-) | 2.745 (2.144, 3.515) | <0.0001 | 1.414 (1.099, 1.821) | 0.007 | 1.541 (1.200, 1.980) | <0.001 | 1.386 (1.073, 1.791) | 0.012 |
| High SII and HBsAg (+) | 1.368 (0.982, 1.906) | 0.064 | 1.527 (1.120, 2.083) | 0.007 | 1.552 (1.152, 2.091) | 0.004 | 1.448 (1.068, 1.964) | 0.017 |
| p for trend(character2integer) | | <0.0001 | | <0.0001 | | <0.0001 | | <0.001 |

Group_hbv_SII

Crudel model: group_hbv_SII

Model 1: group_hbv_SII, age, sex

Model 2: group_hbv_SII, age, sex, eth1, Family_income, edu, marital

Model 3: group_hbv_SII, age, sex, eth1, marital, Family_income, edu, BMI, total_cholesterol, LDL, HDL, TG, Albumin, CKD_EPI_Scr_2009, smoke, alcohol.user, CKD, Hypertension, COPD

The results of this study have several clinical implications. First, SII could serve as a useful biomarker for identifying individuals at higher risk of mortality, particularly in populations with high levels of inflammation [28]. Second, while HBV vaccination remains a crucial public health measure [29], our findings suggest that the protective effects of HBV surface antibody positivity on mortality may be limited in the presence of other risk factors, such as inflammation and comorbidities. Therefore, a more comprehensive approach to risk stratification, incorporating both inflammatory markers and immune status, may be necessary for better predicting mortality risks.

This study has several limitations. First, the observational design prevents us from establishing causality between SII, HBV surface antibody status, and mortality outcomes. Second, while we controlled for a wide range of covariates, residual confounding may still exist. Additionally, the use of NHANES data, while comprehensive, may not fully capture all relevant factors influencing mortality, such as detailed medical histories or specific treatment interventions. Moreover, the NHANES dataset does not distinguish between vaccine- and infection-induced HBsAb positivity, which may obscure differences in inflammatory status and mortality risk related to the underlying source of immunity. Lastly, while we adjusted for key variables in statistical models, potential biases related to missing data or unmeasured confounders cannot be fully excluded.

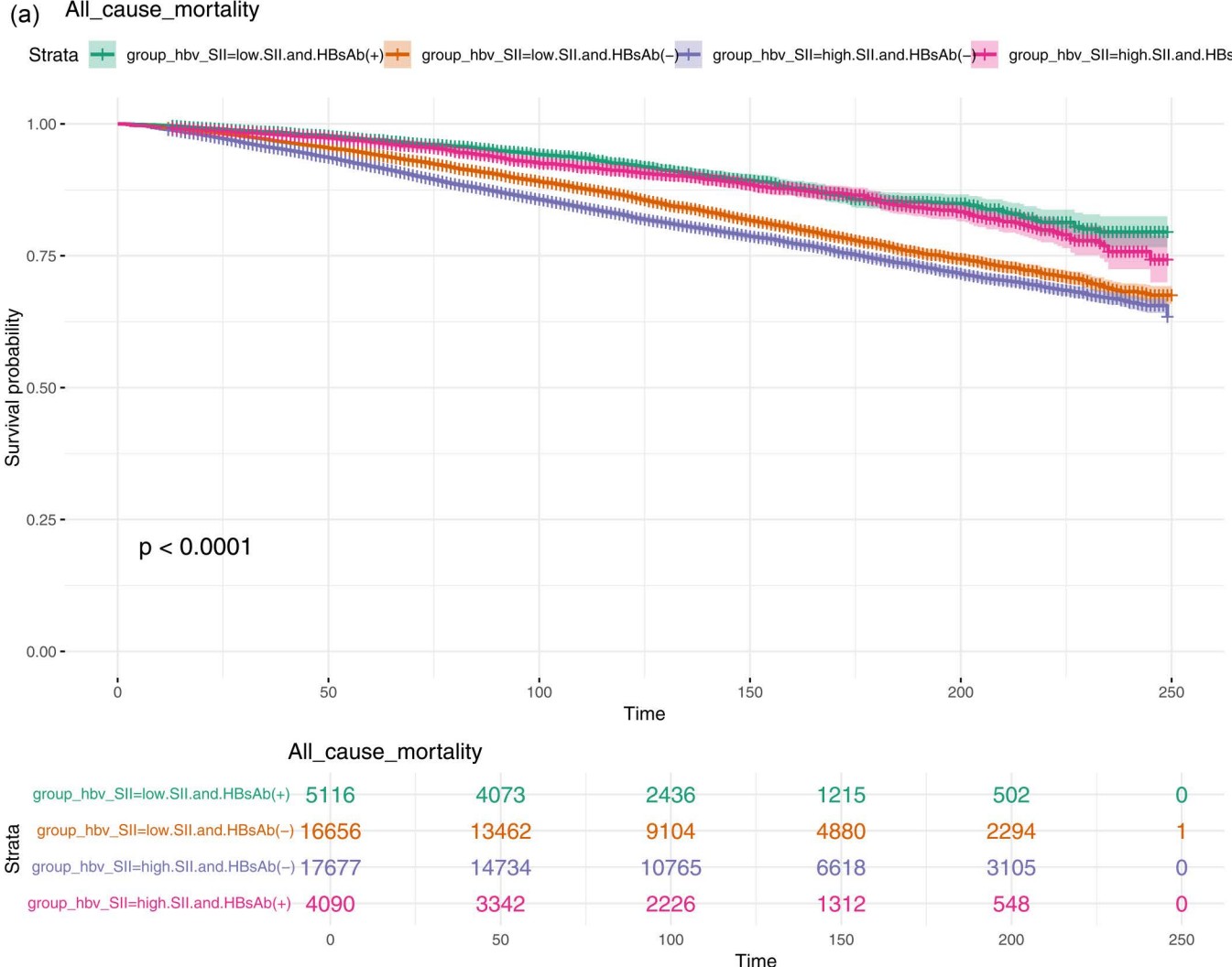

(a) All_cause_mortality

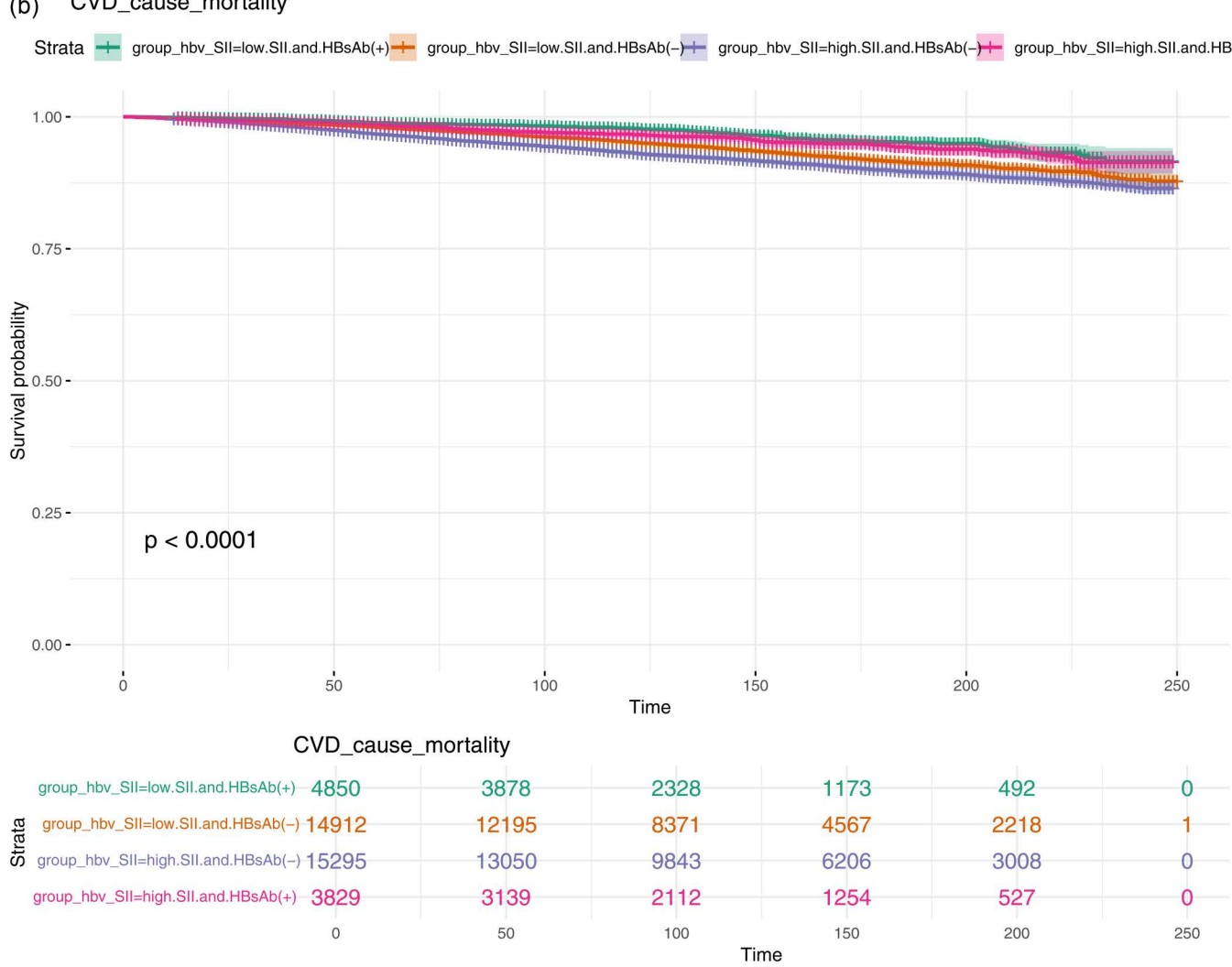

(b) CVD_cause_mortality

Future studies should explore the mechanistic pathways linking inflammation, immune status, and mortality [30]. Longitudinal studies that track changes in SII and HBV antibody status over time may help clarify their dynamic roles in predicting long-term health outcomes. Moreover, research into interventions aimed at reducing inflammation, such as lifestyle modifications or pharmacological treatments, could provide promising strategies for mitigating the mortality risks associated with elevated SII levels.

## 5. Conclusions

Our study highlights the significant role of the Systemic Immune-Inflammation Index (SII) and hepatitis B antibody status in predicting mortality risk. Elevated SII, in combination with negative HBV antibody status, indicates a higher risk for poor health outcomes, including increased all-cause and cardiovascular mortality. On the other hand, a positive HBV antibody status seems to mitigate some of the negative effects of systemic inflammation. These findings underscore the importance of considering both immune and inflammatory markers when assessing long-term health risks, offering potential for improved patient risk stratification and targeted interventions.

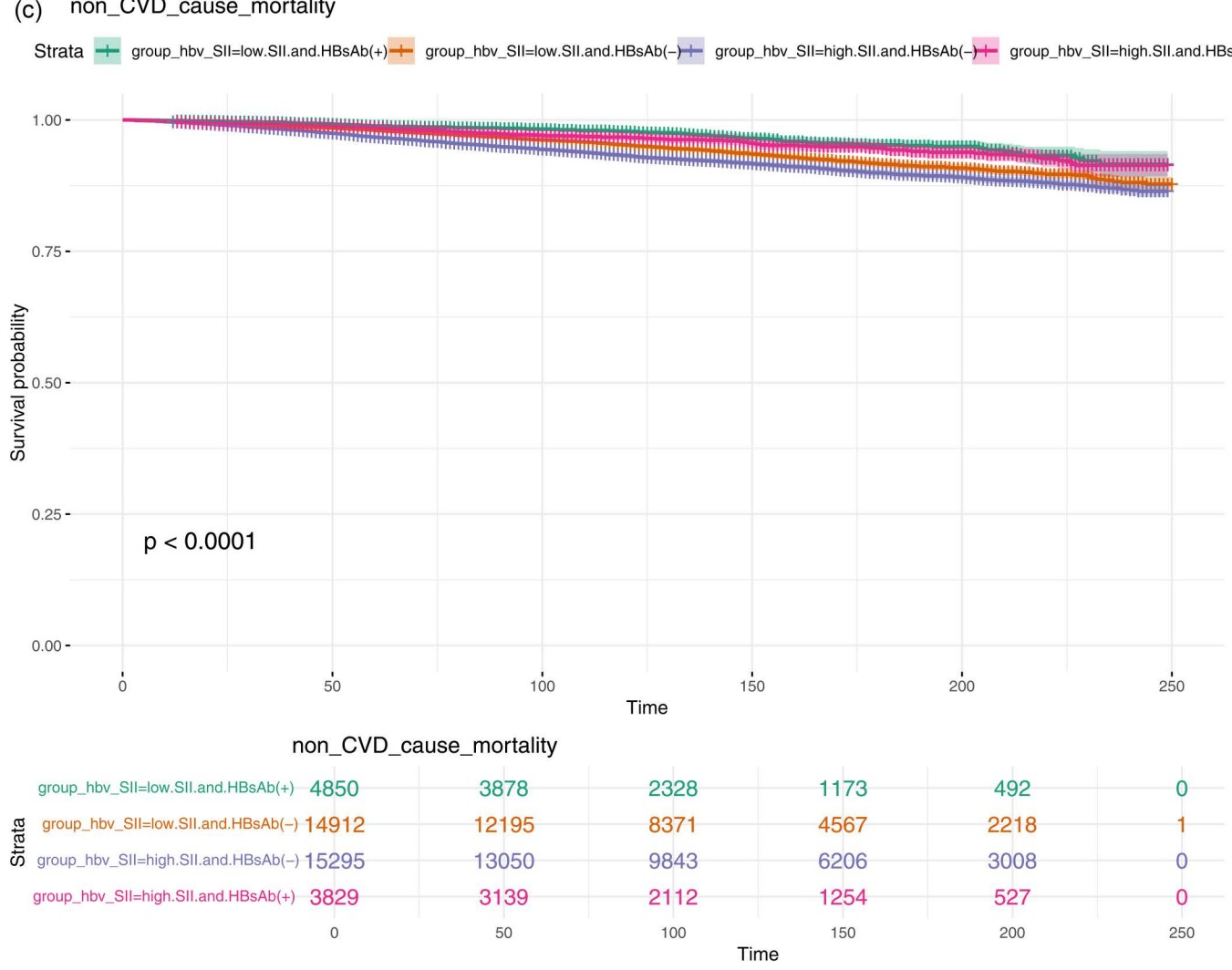

**Fig 3. (a)** All-cause mortality associated with combined systemic immune-inflammation index (SII) and hepatitis B surface antibody (HBsAb) status groups; **(b)** Cardiovascular disease (CVD) mortality associated with combined systemic immune-inflammation index (SII) and hepatitis B surface antibody (HBsAb) status groups; **(c)** Non-cardiovascular disease (non-CVD) mortality associated with combined systemic immune-inflammation index (SII) and hepatitis B surface antibody (HBsAb) status groups.

## Author contributions

**Conceptualization:** Di Zeng, Shaofeng Wang, Jiong Lu.

**Data curation:** Di Zeng, Shaofeng Wang, Bei Li, Jiong Lu.

**Formal analysis:** Di Zeng.

**Investigation:** Nansheng Cheng, Xianze Xiong.

**Methodology:** Nansheng Cheng.

**Software:** Di Zeng, Bei Li, Xianze Xiong.

**Supervision:** Nansheng Cheng, Bei Li, Xianze Xiong.

**Validation:** Di Zeng, Shaofeng Wang, Bei Li, Jiong Lu.

**Visualization:** Di Zeng, Bei Li, Xianze Xiong, Jiong Lu.

**Writing – original draft:** Di Zeng, Shaofeng Wang.

**Writing – review & editing:** Nansheng Cheng, Bei Li, Xianze Xiong, Jiong Lu.

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
