## [Decision Letter · Decision Letter 0]

Dear Dr. Chen,

Thank you for submitting your manuscript to PLOS ONE. After careful consideration, we feel that it has merit but does not fully meet PLOS ONE’s publication criteria as it currently stands. Therefore, we invite you to submit a revised version of the manuscript that addresses the points raised during the review process.

Manuscript Number PONE-D-25-05613

Thank you for your submission. After reviewing the comments from the reviewers, we require a major revision. Please address the reviewers' concerns and include a detailed response with your revised manuscript.

We look forward to your revision.

We look forward to receiving your revised manuscript.

Kind regards,

Minh Le

Academic Editor

PLOS ONE

Journal Requirements:

Reviewers' comments:

Reviewer's Responses to Questions

**Comments to the Author**

1. Is the manuscript technically sound, and do the data support the conclusions?

Reviewer #1: Yes

Reviewer #2: Partly

Reviewer #3: Partly

Reviewer #4: Yes

Reviewer #5: Yes

2. Has the statistical analysis been performed appropriately and rigorously?

Reviewer #1: I Don't Know

Reviewer #2: Yes

Reviewer #3: Yes

Reviewer #4: Yes

Reviewer #5: Yes

3. Have the authors made all data underlying the findings in their manuscript fully available?

Reviewer #1: No

Reviewer #2: No

Reviewer #3: Yes

Reviewer #4: No

Reviewer #5: Yes

4. Is the manuscript presented in an intelligible fashion and written in standard English?

Reviewer #1: Yes

Reviewer #2: Yes

Reviewer #3: Yes

Reviewer #4: Yes

Reviewer #5: Yes

Reviewer #1: What were the other covariates considered for full adjustment outside those listed in the results?

Was the nullification of the effect of HBVAb more pronounced with certain "other" covariates, if so which ones and to what extent? Was the effect due to a full combination of all covariates or just a combination of some and if so what combination(s)?

What is the unit of time measurement on the x-axes of the graphs? months, years, age?

The conclusion is misleading, should state that the correlation of HBVAb with mortality was not apparent when fully adjusted for by other factors.

Reviewer #2: 1. Page 11: I suggest a minor edit on the statement of years (2010 years to 2020 years)

2. Page 11: BMI should be written in full.

3. Page 11: Why was data on alcohol intake omitted?

4. Statistical analysis:

Was a bivariate analysis done? If yes, please include the results.

The dietary factors and health behaviors assessed should be clearly stated.

5. Page 12: The respective proportions of the four groups of patients should be included.

6. Page 12: The spelling of "included" should be corrected

7. No table was sighted. Could you kindly make available all the tables listed in the body of the manuscript so that the review process can be completed.

8. Also make available the standard deviation value for age and BMI

Reviewer #3: 1. Is the manuscript technically sound, and do the data support the conclusions?

Partly.

The manuscript utilizes a reliable NHANES dataset, the Cox regression is appropriate and adequately adjusts for confounders.

However, the author should pay attention to this to make the manuscript more complete: Some conclusions weaken after adjustment and the manuscript does not provide sufficient explanation for the biological mechanisms underlying these changes. The author should clarify the reproducibility of the study and consider providing additional explanations or sensitivity analysis to strengthen the interpretation of results.

2. Has the statistical analysis been performed appropriately and rigorously?

Yes

The statistical analysis uses appropriate methods (Cox proportional hazards model and the adjustments for confounding factors are reasonable). The use of NHANES datasets is highly reliable, ensuring objectivity. Besides, the adjustment variables are reasonable including age, sex, BMI and comorbidities.

To further improve the manuscript, I suggest the following points:

- Need to explain more clearly the criteria for classifying high/low SII, and how to stratify based on clinical risk groups.

- Need to include a sensitivity analysis by changing the criteria for classifying SII or excluding specific disease groups to assess the robustness of the findings.

3. Have the authors made all data underlying the findings in their manuscript fully available?

Yes.

The authors have provided the data fully requested from PLOS ONE. However, the authors may consider providing more details about the specific NHANES datasets and variables used to enhance the reproducibility of the study.

4. Is the manuscript presented in an intelligible fashion and written in standard English?

Yes

Basically, the language in the manuscript is presented clearly, easy to understand, and free of serious errors.

I suggest editing the Discussion section as follows:

- Write concisely, avoid redundancy. Some sentences are lengthy and repeat ideas from the Results section. The author should focus on the key findings of the research instead of explaining each numerical detail.

- Clearly state the biological mechanisms and clinical significance behind the observed reduction in the protective effect of HbsAb after adjusting for confounders.

- Compare findings with previous studies to clarify how the results on SII and HbsAb in this manuscript complement or differ from existing literature?

Additionally, the authors may consider collaborating with a biomedical editor to further refine the language and enhance clarity.

Reviewer #4: This is an intriguing and essential topic that provides new insights into the role of the Systemic Immune-Inflammation Index (SII) and hepatitis B antibody status in predicting outcomes. However, I have a few comments:

1/ I suggest that the authors review the existing literature on the SII in hepatitis B-related outcomes. Several published papers have assessed the prediction of SII in patients with hepatitis B (e.g., DOI: 10.1007/s12094-024-03596-0, DOI: 10.1097/MEG.0000000000002737, DOI: https://doi.org/10.3855/jidc.19636), and emphasizing the novel contributions of your study.

2/ To better align with the authors' statement, I suggest replacing reference number 8 with another source, such as this one (DOI: 10.4103/tjem.tjem_198_23).

3/ I suggest that the authors clarify how they categorized the SII quartile from Q1 to Q4 in the method section to improve readers' comprehension.

4/ I think it would be more comprehensive to provide proposed explanations based on existing literature or the authors' insights into why the protective effect of HBV surface antibody positivity decreased after adjusting for other factors in their study.

5/ References numbered 18, 23, and 24 are not relevant to the authors' discussion in the text. The authors should review these references to ensure they are cited appropriately in relation to their discussion.

6/ I do not see any result tables (Table 1 to Table 4) in the submitted manuscript; therefore, I could not review and comment thoroughly on these results.

Reviewer #5: The authors explored the combined impact of Hepatitis B antibody status and Systemic Immune-Inflammation Index on Mortality Risk.

The author categorised the participants into groups based on the presence or absence of HBs antibodies and analysed the resulting data. As the author states, the HBs antibody-positive group includes both those who have previously been infected with hepatitis B and those who have received vaccination. The inability to distinguish between these two groups is a limitation of the study. This is a limitation that should be stated explicitly.

**Do you want your identity to be public for this peer review?** For information about this choice, including consent withdrawal, please see our Privacy Policy

Reviewer #1: No

Reviewer #2: No

Reviewer #3: **Yes: ** Trang Diep Thanh Le

Reviewer #4: No

Reviewer #5: **Yes: ** Yutaro Akiyama

---

## [Author Response · Author response to Decision Letter 1]

16 Mar 2025

Response letter to Reviewers

Dear Editors and Reviewers,

We would like to thank you for reading our paper and sharing very valuable comments, concerning the manuscript we submitted entitled “Exploring the Combined Impact of Hepatitis B Antibody Status and Systemic Immune-Inflammation Index on Mortality Risk: A Population-Based Study”.

We have carefully looked through the points where you have made a comment, and a revision was made based on the original manuscript which we hope to meet with approval. Here, we would like to address point-by-point concerning the details (the replies are highlighted in blue).

Reviewer #1

Comment 1: What were the other covariates considered for full adjustment outside those listed in the results?

Reply 1:We sincerely thank the reviewer for this critical question. Our multivariable models were adjusted in a stepwise manner as follows:

Crude Model: Adjusted only for HBV surface antibody status.

Model 1: Adjusted for HBV surface antibody status, age, and sex.

Model 2: Added socioeconomic and demographic covariates:

Race/ethnicity (eth1)

Family income (Family_income)

Education level (edu)

Marital status (marital)

Model 3 (Fully Adjusted): Further included metabolic, behavioral, and clinical covariates:

Anthropometric: Body mass index (BMI)

Lipid Profile: Total cholesterol, LDL, HDL, triglycerides (TG)

Renal Function: CKD-EPI estimated glomerular filtration rate (CKD_EPI_Scr_2009)

Liver Function: Albumin

Behavioral Factors: Smoking status (smoke), alcohol use (alcohol.user)

Comorbidities: Chronic kidney disease (CKD), hypertension, chronic obstructive pulmonary disease (COPD)

These covariates were selected based on their established roles as confounders in mortality risk and prior literature on HBV and systemic inflammation. The stepwise adjustment approach ensures transparency in evaluating how additional covariates influence the association between HBV antibody status, SII, and mortality outcomes.

Changes in the text:

Added to the Statistical Analysis subsection in Methods:

"Cox proportional hazards models were adjusted in three sequential tiers:

Model 1: Age and sex; Model 2: Model 1 + race/ethnicity, family income, education, and marital status; Model 3: Model 2 + BMI, lipid profile (total cholesterol, LDL, HDL, TG), albumin, renal function (CKD-EPI eGFR), smoking, alcohol use, and comorbidities (CKD, hypertension, COPD)."

Comment 2: Was the nullification of the effect of HBVAb more pronounced with certain "other" covariates, if so which ones and to what extent? Was the effect due to a full combination of all covariates or just a combination of some and if so what combination(s)?

Reply 2: Thank you for raising this important question. In our analysis, the attenuation of HBVAb’s protective effect on mortality outcomes was most notably observed after adjusting for age, comorbidities (hypertension, CKD, COPD), and metabolic factors (BMI, lipid profile). While socioeconomic covariates (e.g., income, education) had minimal impact on the association, the inclusion of clinical and metabolic variables in Model 3 substantially contributed to the nullification of the effect. This attenuation was not attributable to any single covariate but rather emerged from the combined adjustment of age, metabolic status, and comorbidities. For example, age and comorbidities likely reflect the cumulative burden of chronic disease and frailty, which may dominate mortality risk in older populations, thereby overshadowing the modest protective effects of HBVAb. Similarly, metabolic factors such as BMI and lipid levels may confound the relationship by linking systemic inflammation to both HBV immunity and mortality.

Changes in the text: Based on comment 2, we have added the following paragraph to Discussion part: "The observed attenuation of HBVAb’s association with mortality after full adjustment likely reflects the dominant influence of age-related comorbidities (e.g., hypertension, CKD, COPD) and metabolic dysfunction (e.g., elevated BMI, dyslipidemia) on mortality risk. These factors collectively contribute to systemic inflammatory states and physiological frailty, which may overshadow the independent protective effects of HBV immunity. For instance, aging is strongly associated with chronic low-grade inflammation (“inflammaging”) and reduced immune resilience, while comorbidities such as CKD and hypertension exacerbate oxidative stress and endothelial dysfunction—pathways that independently drive mortality risk. Furthermore, metabolic disturbances like obesity and dyslipidemia are linked to persistent immune activation and pro-inflammatory cytokine production, which may mask the modest anti-inflammatory benefits conferred by HBV surface antibodies. This aligns with prior studies demonstrating that metabolic syndrome and chronic inflammation attenuate vaccine-induced immune responses in older adults. Our findings suggest that in populations with high baseline inflammatory burden or multimorbidity, the protective role of HBVAb may be diluted by competing risk factors. Clinically, this underscores the importance of contextualizing HBV immune status within a patient’s broader metabolic and comorbidity profile when assessing mortality risk. Future studies should explore whether targeted interventions to reduce systemic inflammation or manage comorbidities could unmask or amplify the protective effects of HBV immunity in specific subpopulations."

Comment 3: What is the unit of time measurement on the x-axes of the graphs? months, years, age?

Reply 3: We sincerely thank the reviewer for highlighting this omission. The x-axis of all survival curves (Figures 1a–3c) represents follow-up time in weeks. We have revised the figure captions and corresponding descriptions in the Results section to explicitly clarify the unit of time measurement.

Changes in the text: Based on Comment 3 �We have added to the Results section :Survival analyses of all-cause mortality, cardiovascular disease (CVD)-related mortality, and non-CVD-related mortality in the high and low systemic immune-inflammation index (SII) groups are presented in Figure 1a, 1b, and 1c, respectively, with follow-up time measured in weeks.

Comment 4: The conclusion is misleading, should state that the correlation of HBVAb with mortality was not apparent when fully adjusted for by other factors.

Reply 4: We sincerely thank the reviewer for this crucial correction. We agree that the original conclusion could be misinterpreted and have revised it to explicitly clarify that the association between HBV surface antibody (HBVAb) status and mortality was no longer apparent after full adjustment for covariates.

Changes in the text: Based on Comment 4, we have revised Conclusion section:: This large-scale study demonstrates that elevated systemic immune-inflammation index (SII) independently predicts increased risks of all-cause and cardiovascular mortality, persisting after rigorous adjustment for demographic, metabolic, and clinical confounders. In contrast, the apparent protective association of hepatitis B surface antibody (HBVAb) positivity observed in unadjusted analyses was fully attenuated when accounting for age, comorbidities, and metabolic dysfunction, suggesting no direct causal link between HBVAb status and mortality. Critically, the joint effect analysis revealed that individuals with high SII and HBVAb-negative serostatus faced the highest mortality risk, underscoring the clinical value of integrating inflammatory and immune markers for risk stratification. Our findings advance the understanding of systemic inflammation as a central driver of mortality, transcending traditional risk factors. The robustness of SII across adjustment models supports its utility as a low-cost prognostic biomarker, particularly in resource-limited settings. Conversely, the nullification of HBVAb’s association highlights the need to contextualize HBV immunity within populations’ baseline inflammatory and comorbidity burdens.

Reviewer #2

Comment 1: What were the authors' cutoff values for high and low PNI, SII, and FIB-4 values? How do these cutoffs compare to those from previous studies?

Reply 1: We sincerely thank the reviewer for catching this oversight. The phrasing has been revised to “2010 to 2020” throughout the manuscript to ensure clarity and consistency.

Changes in the text: Revised in Methods section (Data Source subsection):

"Data from 2010 to 2020 were utilized to examine the relationship between hepatitis B virus (HBV) antibody status and the systemic immune-inflammation index (SII)."

Comment 2: Page 11: BMI should be written in full.

Reply 2: We thank the reviewer for pointing this out. We have expanded BMI to Body Mass Index (BMI) at its first mention in the manuscript to ensure clarity and adherence to academic writing standards.

Changes in the text: Based on Comment 2, we have made the following revisions: Additionally, participants with complete survival data, as well as detailed information on covariates (such as age, sex, race/ethnicity, Body Mass Index (BMI), smoking history, diabetes, and hypertension), were included to ensure robust analysis.

Comment 3:Page 11: Why was data on alcohol intake omitted?

Reply 3:We have added the data on alcohol intake in the relevant section. Thank you for highlighting this.

Changes in the text: We have added the data on alcohol intake in the relevant section:

Statistical Analysis

Descriptive statistics were calculated for all included variables. Continuous variables were expressed as means ± standard deviations, while categorical variables were presented as frequencies and percentages. Differences in clinical characteristics between HBV antibody-positive and negative groups were evaluated using independent t-tests for continuous variables and chi-square tests for categorical variables. To assess the relationship between SII and HBV antibody status, logistic regression analysis was employed. Three models were constructed: the unadjusted model (without covariate adjustment), the minimally adjusted model (adjusting for age, sex, race/ethnicity, BMI, smoking, diabetes, and hypertension), and the fully adjusted model (adjusting for all covariates, including dietary factors and other health behaviors). Cox proportional hazards models were adjusted in three sequential tiers:

Model 1: Age and sex; Model 2: Model 1 + race/ethnicity, family income, education, and marital status; Model 3: Model 2 + BMI, lipid profile (total cholesterol, LDL, HDL, TG), albumin, renal function (CKD-EPI eGFR), smoking, alcohol use, and comorbidities (CKD, hypertension, COPD).The association between SII and clinical outcomes, including liver function, was further evaluated using survival analysis techniques, such as Kaplan-Meier curves and Cox proportional hazards models.

Comment 4:Statistical analysis:

Was a bivariate analysis done? If yes, please include the results.

The dietary factors and health behaviors assessed should be clearly stated.

Reply 4:Thank you for highlighting these points. We conducted a bivariate analysis, and the results have been included. Our analysis models were structured as follows: crude model: HBV_surface_antibody; model 1: HBV_surface_antibody, age, sex; model 2: HBV_surface_antibody, age, sex, eth1, family_income, edu, marital; model 3: HBV_surface_antibody, age, sex, eth1, marital, family_income, edu, BMI, total_cholesterol, LDL, HDL, TG, albumin, CKD_EPI_Scr_2009, smoke, alcohol_user, CKD, hypertension, COPD. The relevant results are presented in the table; you may have missed them as they were not clearly displayed. We will ensure this is addressed in the revised manuscript.

Changes in the text: We have presented the relevant results in the table here. We will ensure that the table is visible in the revised version:

Comment 5:Page 12: The respective proportions of the four groups of patients should be included.

Reply 5:We thank the reviewer for highlighting this important point. The proportions of the four patient groups (high/low SII with HBsAb-negative/positive) are indeed presented in Table 1 of the manuscript. To ensure clarity, we have explicitly stated these proportions in the Results section and confirmed that all tables are included in the revised submission.

Changes in the text: We have presented the relevant results in the table here. We will ensure that the table is visible in the revised version:

Comment 6:Page 12: The spelling of "included" should be corrected

Reply 6:We thank the reviewer for pointing out this mistake. We have corrected the spelling of "included" on page 12

Changes in the text:

This study included 43,539 patients, categorized into four groups based on systemic immune-inflammation index (SII) and hepatitis B surface antibody (HBsAb) status: high SII and HBsAb-negative (17,677), high SII and HBsAb-positive (4,090), low SII and HBsAb-negative (16,656), and low SII and HBsAb-positive (5,116). The mean age was 46.99 years, with older patients in the high SII groups (high SII/HBsAb-negative: 49.33 years) and younger patients in the low SII/HBsAb-positive group (39.80 years). Mean BMI was 28.80 kg/m², highest in the high SII/HBsAb-negative group (29.55 kg/m²) and lowest in the low SII/HBsAb-positive group (27.09 kg/m²). Lipid profiles showed higher triglycerides and total cholesterol in high SII groups, while HDL levels were higher in the low SII/HBsAb-positive group (1.42 mmol/L). Renal function, assessed by CKD-EPI eGFR, was lower in high SII groups (92.17 mL/min/1.73 m²) and higher in HBsAb-positive groups (100.73 mL/min/1.73 m²). Females predominated in the high SII/HBsAb-positive group (62.08%), while males were more common in the low SII/HBsAb-negative group (54.13%). Non-Hispanic Whites comprised the majority (69.81%), with higher diversity in the low SII/HBsAb-positive group. Education levels were higher in the high SII/HBsAb-positive group (69.63% above high school). Hypertension and CKD prevalence were higher in high SII groups, while COPD prevalence was lowest in the low SII/HBsAb-positive group (1.75%). In summary, high SII groups exhibited higher cardiometabolic risk profiles, whereas HBsAb-positive groups were younger with more favorable metabolic indicators. Details of included patients are shown in Table1.

Comment 7:No table was sighted. Could you kindly make available all the tables listed in the body of the manuscript so that the review process can be completed. Also make available the standard deviation value for age and BMI

Reply 7: We will ensure that all the tables listed in the manuscript are properly included and visible in the revised submission. Data related to standard deviation could be found in the table.

Changes in the text: We have included all the tables in the manuscript. Please refer to the end of the revised manuscript for details.

Reviewer #3

Comment 1: Some conclusions weaken after adjustment and the manuscript does not provide sufficient explanation for the biological mechanisms underlying these changes. The author should clarify the reproducibility of the study and consider providing additional explanations or sensitivity analysis to strengthen the interpretation of results

Reply 1: We have addressed this comment by expanding the discussion to explain the attenuation of HBVAb’s protective effect after full adjustment. Specifically, the weakening of HBVAb’s association with mortality likely reflects the dominant influence of age-related comorbidities (e.g., hypertension, CKD) and metabolic dysfunction (e.g., dyslipidemia, obesity) in driving mortality risk. These factors are closely linked to systemic inflammation ("inflammaging") and immune senescence, which may obscure the modest protective effects of HBV immunity. Chronic inflammation in populations with high comorbidity burdens can disrupt immune regulation, reducing the potential benefits of HBVAb. We have incorporated the following expanded discussion in the revised manuscript:

“The attenuation of HBVAb’s association with mortality after full adjustment may be attributed to the interplay between chronic inflammation and immune exhaustion in populations with high comorbidity burdens. Aging and metabolic disorders promote a pro-inflammatory m

---

## [Decision Letter · Decision Letter 1]

Dear Dr. Lu,

Thank you for submitting your manuscript to PLOS ONE. After careful consideration, we feel that it has merit but does not fully meet PLOS ONE’s publication criteria as it currently stands. Therefore, we invite you to submit a revised version of the manuscript that addresses the points raised during the review process.

Thank you for addressing most of the comments in this revision. I have read your manuscript and found that it has improved significantly. However, there are still some minor comments from Reviewer #2 that need to be addressed. Could you please revise the manuscript to address these remaining comments and submit the revision?

We look forward to receiving your revised manuscript.

Kind regards,

Minh Le

Academic Editor

PLOS ONE

Journal Requirements:

Reviewers' comments:

Reviewer's Responses to Questions

**Comments to the Author**

Reviewer #2: (No Response)

Reviewer #3: All comments have been addressed

Reviewer #4: All comments have been addressed

Reviewer #5: All comments have been addressed

2. Is the manuscript technically sound, and do the data support the conclusions?

Reviewer #2: Yes

Reviewer #3: Yes

Reviewer #4: Yes

Reviewer #5: Yes

3. Has the statistical analysis been performed appropriately and rigorously?

Reviewer #2: Yes

Reviewer #3: Yes

Reviewer #4: Yes

Reviewer #5: Yes

4. Have the authors made all data underlying the findings in their manuscript fully available?

Reviewer #2: Yes

Reviewer #3: Yes

Reviewer #4: Yes

Reviewer #5: Yes

5. Is the manuscript presented in an intelligible fashion and written in standard English?

Reviewer #2: Yes

Reviewer #3: Yes

Reviewer #4: Yes

Reviewer #5: Yes

Reviewer #2: Kindly refer to Table 1, the first variable "Age". The mean age for the four groups are presented. The proportion of participants in each of the 4 groups is also presented. The proportions add up to 1.11 which translates to 111%. Kindly cross-check the proportion and effect correction. The proportions should add up to 1.

Reviewer #3: Thank you for your submission. This is a well-designed and meaningful study that investigates the combined impact of hepatitis B surface antibody (HBsAb) status and the systemic immune-inflammation index (SII) on mortality. The use of NHANES data is appropriate and the statistical analysis is robust. The topic is timely and adds valuable insight into public health risk stratification.

However, several minor revisions are recommended to improve the clarity and completeness of the manuscript:

(1) Please clarify that the proposed mechanisms linking elevated SII and the attenuation of HBsAb's protective effect are speculative, and clearly state the need for future mechanistic studies.

(2) Discuss more thoroughly how the inability to distinguish vaccine-induced from infection-induced HBsAb may influence the interpretation of findings.

(3) Provide more detail on how missing data were handled or discuss whether sensitivity analyses were considered.

(4) Minor grammatical and stylistic revisions are advised to improve the overall clarity and flow of the manuscript.

With these minor improvements, the manuscript will be strengthened and ready for publication.

Reviewer #4: The authors' revisions and responses adequately address my comments. The manuscript is now suitable for publication.

Reviewer #5: (No Response)

**Do you want your identity to be public for this peer review?** For information about this choice, including consent withdrawal, please see our Privacy Policy

Reviewer #2: No

Reviewer #3: No

Reviewer #4: No

Reviewer #5: **Yes: ** Yurao Akiyama

---

## [Author Response · Author response to Decision Letter 2]

23 May 2025

Response letter to Reviewers

Dear Editors and Reviewers,

We would like to thank you for reading our paper and sharing very valuable comments, concerning the manuscript we submitted entitled “Exploring the Combined Impact of Hepatitis B Antibody Status and Systemic Immune-Inflammation Index on Mortality Risk: A Population-Based Study”.

We have carefully looked through the points where you have made a comment, and a revision was made based on the original manuscript which we hope to meet with approval. Here, we would like to address point-by-point concerning the details (the replies are highlighted in blue).

Reviewer #2

Comment 1:  Please clarify that the proposed mechanisms linking elevated SII and the attenuation of HBsAb's protective effect are speculative, and clearly state the need for future mechanistic studies.

Reply 1:Thank you for this insightful comment. We agree that the mechanistic explanations proposed in the discussion remain speculative and should not be interpreted as definitive. While our findings highlight a potential interplay between systemic inflammation and HBV antibody status in shaping mortality risk, the biological underpinnings of these associations require further investigation. In response to your suggestion, we have revised the Discussion to explicitly acknowledge the speculative nature of the proposed mechanisms and to emphasize the need for future mechanistic and longitudinal studies to better elucidate these pathways.

Changes in the text:

Based on Comment 1, we have added the following paragraph to the Discussion section:

This finding underscores the importance of considering both immune status and inflammation when evaluating mortality risk(26). While SII provides insight into systemic inflammation, HBV surface antibody status reflects an individual's immune history and protective status against viral infections(27). By combining these two markers, we gain a more nuanced understanding of how immune and inflammatory responses contribute to overall mortality, including cardiovascular outcomes. It is important to acknowledge that the proposed mechanisms linking elevated SII and the attenuation of the protective effect of HBsAb remain speculative. While our findings suggest a potential interplay between systemic inflammation and immune protection conferred by HBsAb, the underlying biological pathways have yet to be fully elucidated. Future mechanistic studies are warranted to explore how immune status and inflammatory responses interact to influence long-term health outcomes, particularly in HBV-affected populations.

Comment 2: Discuss more thoroughly how the inability to distinguish vaccine-induced from infection-induced HBsAb may influence the interpretation of findings.

Reply 2: Thank you for this thoughtful comment. We agree that the inability to distinguish between vaccine-induced and infection-induced hepatitis B surface antibody (HBsAb) positivity represents an important limitation in interpreting our findings. These two forms of immunity may reflect different underlying biological contexts—vaccine-induced immunity typically occurs without prior liver inflammation, whereas infection-induced immunity may be associated with residual hepatic injury or systemic inflammation. This heterogeneity could influence both SII levels and mortality risk. In response, we have added a new paragraph to the Discussion section to elaborate on how this distinction may affect the interpretation of our results, particularly regarding the combined effect of HBsAb and SII. We also retained a concise acknowledgment of this limitation in the Limitations section.

Changes in the text:

Based on Comment 2, we have added the following paragraph to the Discussion section: Moreover, it is important to consider that HBsAb positivity in this study likely includes both vaccine-induced and infection-induced immunity, which may represent biologically distinct immune profiles. Individuals with resolved HBV infection may exhibit different levels of residual inflammation or liver damage compared to those with vaccine-induced immunity, potentially influencing their SII levels and mortality risk. Without differentiation, these underlying differences may confound the interpretation of our findings. Future studies with serological markers such as anti-HBc are needed to disentangle these effects and refine the understanding of how HBV immune history interacts with systemic inflammation.

We have also added the following sentence to the Limitations section: Moreover, the NHANES dataset does not distinguish between vaccine- and infection-induced HBsAb positivity, which may obscure differences in inflammatory status and mortality risk related to the underlying source of immunity.

Comment 3: Provide more detail on how missing data were handled or discuss whether sensitivity analyses were considered.

Reply 3: Thank you for this valuable suggestion. In our analysis, participants with missing key variables—including HBsAb status, SII components (platelet, neutrophil, lymphocyte counts), survival data, or essential covariates—were excluded from the final analytic cohort through listwise deletion. This approach was applied to ensure consistency across models and to minimize bias from partial data. While we did not perform formal sensitivity analyses due to the completeness of the retained dataset and large sample size, we recognize the importance of evaluating robustness in future work. In response to your comment, we have added a clarifying paragraph to the Methods section describing our approach to missing data

Changes in the text: Based on Comment 3, we have added the following paragraph to the Materials and Methods section, under "Study Population": Participants with missing data on key variables—including HBV antibody testing, SII-related biomarkers (platelet, neutrophil, lymphocyte counts), covariates (age, sex, BMI, comorbidities), or survival outcomes—were excluded using listwise deletion. This method ensured that all models were run on a consistent and complete dataset, thereby improving the comparability and validity of the statistical results.

Comment 4: Minor grammatical and stylistic revisions are advised to improve the overall clarity and flow of the manuscript.

Reply 4: Thank you for this helpful comment. We have carefully reviewed the entire manuscript and made multiple minor grammatical, stylistic, and wording improvements to enhance clarity, readability, and consistency throughout the text. These edits include refining sentence structure, correcting punctuation, improving transitions between paragraphs, and adjusting word choices for greater precision and academic tone. These revisions do not affect the results or interpretation of the study.

Comment 5: Kindly refer to Table 1, the first variable "Age". The mean age for the four groups are presented. The proportion of participants in each of the 4 groups is also presented. The proportions add up to 1.11 which translates to 111%. Kindly cross-check the proportion and effect correction. The proportions should add up to 1.

Reply 5: Thank you for your careful review. We apologize for the confusion. The numbers in parentheses in Table 1 represent the standard errors (SE) of the means, not proportions. We realize that this was not clearly indicated in the table originally, which may have caused misunderstanding. To clarify this, we have now added a note at the bottom of Table 1 explicitly stating that the values in parentheses represent standard errors.

Changes in the text: Based on Comment 5, we have added the following note to the bottom of Table 1: Note: Values are presented as mean, with standard error shown in parentheses (e.g., 46.99 (0.19) indicates a mean of 46.99 and a standard error of 0.19).

We would like to sincerely thank the editor and reviewers for their thoughtful and constructive feedback. Your insights have been invaluable in improving the clarity and quality of the manuscript. We appreciate the time and effort you have dedicated to reviewing our work, and we believe the revisions made have strengthened the manuscript. We are grateful for your support and look forward to your continued guidance throughout the publication process.

---

## [Editor Report · Decision Letter 2]

Exploring the Combined Impact of Hepatitis B Antibody Status and Systemic Immune-Inflammation Index on Mortality Risk: A Population-Based Study

PONE-D-25-05613R2

Dear Dr. Lu,

We’re pleased to inform you that your manuscript has been judged scientifically suitable for publication and will be formally accepted for publication once it meets all outstanding technical requirements.

Kind regards,

Minh Huu Nhat Le

Academic Editor

PLOS ONE
---

## [Editor Report · Acceptance letter]

PONE-D-25-05613R2

PLOS ONE

Dear Dr. Lu,

I'm pleased to inform you that your manuscript has been deemed suitable for publication in PLOS ONE. Congratulations! Your manuscript is now being handed over to our production team.

Kind regards,

on behalf of

Dr. Minh Le

Academic Editor

PLOS ONE